# T-cell stimuli independently sum to regulate an inherited clonal division fate

J.M. Marchingo[1,2,†], G. Prevedello[3], A. Kan[1,2], S. Heinzel[1,2], P.D. Hodgkin[1,2,*] & K.R. Duffy[3,*]

In the presence of antigen and costimulation, T cells undergo a characteristic response of expansion, cessation and contraction. Previous studies have revealed that population-level reproducibility is a consequence of multiple clones exhibiting considerable disparity in burst size, highlighting the requirement for single-cell information in understanding T-cell fate regulation. Here we show that individual T-cell clones resulting from controlled stimulation *in vitro* are strongly lineage imprinted with highly correlated expansion fates. Progeny from clonal families cease dividing in the same or adjacent generations, with inter-clonal variation producing burst-size diversity. The effects of costimulatory signals on individual clones sum together with stochastic independence; therefore, the net effect across multiple clones produces consistent, but heterogeneous population responses. These data demonstrate that substantial clonal heterogeneity arises through differences in experience of clonal progenitors, either through stochastic antigen interaction or by differences in initial receptor sensitivities.

[1] Division of Immunology, The Walter and Eliza Hall Institute of Medical Research, 1G Royal Parade, Parkville, Victoria 3052, Australia. [2] Department of Medical Biology, The University of Melbourne, Parkville, Victoria 3010, Australia. [3] Hamilton Institute, Maynooth University, Maynooth, Co Kildare W23 WK26, Ireland. † Present address: Division of Cell Signalling and Immunology, School of Life Sciences, University of Dundee, Dundee DD1 5EH, UK. * These authors contributed equally to this work. Correspondence and requests for materials should be addressed to P.D.H. (email: hodgkin@wehi.edu.au) or to K.R.D. (email: ken.duffy@nuim.ie).

T-cell immunity against infection requires the activation and expansion of a small number of pathogen-specific cells to form a larger pool of protective lymphocytes[1]. The net behaviour of these rare pathogen-specific clones dictates the characteristics of the population response and, for a given infection, results in a highly reproducible response magnitude. Despite this consistency in population responses, *in vivo* measurements of clonal burst size and phenotype have revealed substantial heterogeneity between clones[2–7], highlighting the requirement for single-cell information in understanding T-cell fate regulation. From these studies, a critical question arises: how is clonal diversity within the T-cell response generated? In particular, to what extent is variation in clonal outcomes intrinsically inherited from the initial cell and how much arises through deterministic and stochastic processes, both intrinsic and extrinsic, experienced by individual daughter cells after the initial activating events[8]? Here we direct this question to examine the considerable variation in proliferative capacity of individual T cells following stimulation[2–5].

Population-level studies have demonstrated that T cells with identical T-cell receptors (TCRs) respond heterogeneously[9–11] and, even under controlled *in vitro* conditions[9], divide a variable number of times before stopping and reverting to a quiescent state. Following previous studies[9,12,13], we defined the generation in which an activated lymphocyte returns to quiescence to be its division destiny (DD) and asked how heterogeneity in DD is generated at a family level. Figure 1 presents two alternative clonal level possibilities: first, the population distribution of DD (Fig. 1a) could arise through strongly clonally correlated DD fates; and, second, the heterogeneity might emerge from highly discordant family DD histories (Fig. 1b top and bottom panels, respectively). Identifying strong clonal concordance would indicate that DD is a lineage primed, inherited property. In contrast, clonal discordance in DD fate, in which cells stop over multiple generations, could result from deterministic programming through an asymmetric cell division[14,15] or by stochastic regulation[16,17]. Published data cannot distinguish between these possibilities.

Any clonal level answer to the question of relative concordance in DD must also be reconciled with a further striking population level observation: T-cell DD is regulated by the type and the strength of the signals received, and many signal combinations result in both the means and variances of population DD distributions summing linearly, illustrated in Fig. 1c (ref. 9). This observation suggests independence of the effects of signals driving DD. Thus the solution to the familial genesis of DD variation posed in Fig. 1a,b must also address how variable outcomes at single cell level result from fates of clonal family trees (Fig. 1d). Here we sought to identify the source of DD variation, and identify how signal integration that is additive at the population level results from, and is consistent with, clonal family behaviour. To address these questions we develop and utilize a novel multiplex clonal division-tracking assay based on the combinatorial use of multiple division tracking dyes. Using this technique we reveal that CD8+ T-cell clones are imprinted with highly correlated division fates during the early immune response, such that progeny cells from clonal families cease dividing in the same or adjacent generations, with inter-clonal variation producing burst-size diversity. We report a new mathematical framework enabling us to deduce that the summation of T-cell stimuli effects on division number at the clonal level are stochastic and independent. This clonal addition of signals results in reproducible population-level responses, with substantial clonal heterogeneity arising through differences in stochastic antigen interaction and initial receptor sensitivity.

## Results

**A novel multiplex assay to measure clonal division.** To investigate clonal regulation of T-cell DD, we developed a method to determine familial proliferation features of a large number of founder cells. To simultaneously track proliferation of multiple clonal families in a single culture well, we labelled lymphocytes with quantitatively distinct combinations and concentrations of the division tracking dyes 5-(and 6)-carboxyfluorescein diacetate succinimidyl ester (CFSE), CellTrace Violet (CTV) and Cell Proliferation Dye eFluor670 (CPD) (Fig. 2a,b)[18,19]. In this way we created up to 10 fluorescently distinct populations in a single co-culture. To measure clonal regulation of division fate using this multiplex tracking dye method, we adopted the minimal *in vitro* stimulation system that was established previously to determine variation in DD at a population level (as in Fig. 1a)[9]. OT-I CD8+ T cells, which recognize SIINFEKL (N4) peptide presented on H2K$^b$, and deficient for the pro-apoptotic protein Bim (OT-I/$Bcl2l11^{-/-}$) were purified, labelled with the division tracking dye multiplex and stimulated by peptide self-presentation in the presence of anti-mouse IL-2 blocking antibody (clone S4B6; Fig. 2a–c). Bim-deficiency enhanced cell survival without altering DD and addition of anti-mouse IL-2 blocking antibody limited the autocrine IL-2 present in the culture, allowing T cells to reach DD within the range of division tracking dyes[9]. After 26 h (just prior to the first division) cells were sorted so that a single stimulated but undivided cell, identified by high forward scatter (FSC) fluorescence and undiluted division tracking dye, from each fluorescently distinct population was sorted per well. Cells were returned to culture, without peptide but with S4B6, until analysis by flow cytometry at 54, 62 and 72 h post stimulation (Fig. 2d,e), capturing times when most cells were reaching DD without considerable cell death having occurred[9]. Clonal family division fate from each labelling configuration was identified using the fluorescent gating scheme outlined in Fig. 2f. Small cell size was used to indicate return to quiescence, as previously defined by correlation with a cell cycle reporter of $G_0$ (refs 9,12,20; Fig. 2f, also Supplementary Fig. 2, and Methods).

**Assessing synchronicity in clonal T-cell proliferation.** Figure 3a–c and Supplementary Fig. 3 displays results of an application of the multiplex clonal stimulation assay to OT-I/$Bcl2l11^{-/-}$ CD8+ T cells stimulated by N4, αCD28 and IL-2 (added as human IL-2 (hIL-2) to overcome blocking by S4B6). Fig. 3a presents a population control culture where 500–2,000 cells were sorted from each of eight labelled groups, stimulated and harvested at 72 h post stimulation. As described in Fig. 2, the data in Fig. 3a enabled the identification of generation-gates for each dye-combination. Fig. 3b shows two single co-cultured wells harvested at the same time, with the population-determined gates overlaid, allowing the determination of the generation of each individual cell in each clonal family. These data illustrate how the multiplexing system permits distinct families to be isolated and followed in the co-culture, empowering analysis of familial concordance under these, and altered, culture conditions.

**T-cell proliferation is synchronous.** The data shown in Fig. 3c was analysed for features of clonal proliferation synchronicity. From an input of 224 clones across three time points, we detected at least one progeny cell from 171 clones (76% of the input). In 42% of these clones, all progeny cells were measured, which is comparable to the recovery from time-lapse microscopy of non-adherent cells[12]. The proportion of cells per generation from the pooled clonal progeny and population control were comparable (Supplementary Fig. 4).

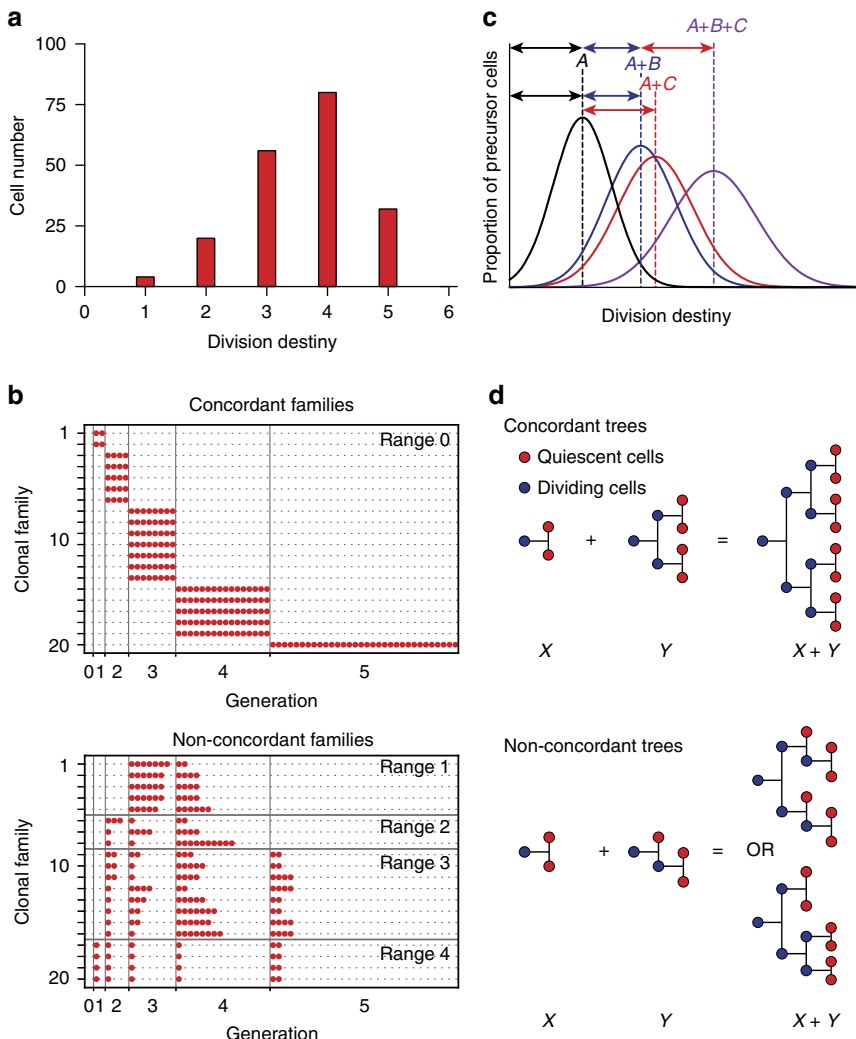

**Figure 1 | How is T-cell division destiny (DD) regulated at a clonal level?** Hypothetical data. (**a**) When apparently identical T cells are stimulated, they proliferate to different extents, resulting in the population of progeny cells returning to quiescence (that is, DD) across multiple generations. (**b**) Two distinct clonal family DD behaviours are consistent with the data in **a**; a highly concordant clonal DD that would arise if DD was inherited (top panel) or a highly discordant family DD (bottom panel), which could occur through stochastic or deterministic regulation. Each row represents a single clone, with circles showing progeny cells reaching DD per generation. Clonal range = maximum − minimum generation number. (**c**) Signals affecting T-cell DD have been shown to add together at the population level[9]. (**d**) If signal effects are independent, clonal family tree addition offers a possible explanation. Addition of concordant trees results in a tree that is also concordant (top panel). Addition of discordant family trees is more complex, as we must allow combinatorial interlacements of tree subsections to represent all possible contributing interactions in time and place (bottom panel, Supplementary Fig. 1 and Methods). Despite the distinct family trees in **d**, lower panel, the numbers of DD cells per generation (red circles) are the same, which is a general property (see Methods).

Clonal T-cell division progression was notably concordant (Fig. 3c, Supplementary Fig. 3). At 54 h a considerable proportion of the clones were still dividing (blue), with more quiescent progeny cells (red) being found at 62 and 72 h. Irrespective of time point or quiescence status, the progeny of an individual clone exhibited strongly synchronous proliferation, with the progeny of 85% of clones being detected in the same generation. In all remaining clones, progeny were detected in adjacent generations (Fig. 3c). This was also observed under conditions when peptide persisted in the culture (Supplementary Fig. 5), and was consistent with previous observations of a high degree of synchrony in T- and B-cell sibling and cousin division times[12,20–22]. In all mixed phenotype clones, the dividing progeny were found in the same or previous generation compared with the quiescent cells (Fig. 3c, Supplementary Fig. 3). This suggested that the discordance observed arose through slight variations in division timing or potential within an otherwise synchronized clone, consistent with Fig. 1b, upper panel.

**Division fate is concordant in response to different stimuli.** For this T-cell system, it has been shown that population level DD is strongly influenced by a range of signals that act alone or in concert to extend the number of mitotic cycles cells undergo[9]. To add to the N4 + αCD28 + IL-2 data presented in Fig. 3, we set up conditions to examine concordance with N4 alone, N4 + αCD28, and N4 + IL-2, added as hIL-2. To determine clonal DD for each condition, we pooled data from 54, 62 and 72 h for families where all detected progeny were quiescent (Fig. 4a, Supplementary Fig. 6a). To summarize the concordance of a clonal family we measured the difference in generation number between the greatest and least DD of any descendant of a clone, a quantity we

called the range. Fig. 4b and Supplementary Fig. 6b plots the distribution of range for each culture condition, illustrating that for the vast majority of families perfect concordance is observed.

**Clonal family DD is concordant**. While superficially the data indicate strong familial features for each stimulatory condition, we sought to quantitatively investigate how much within-family correlation in DD fate was necessary to explain the data in Fig. 4b, taking into account that the experimental method does not sample all cells. To that end, we developed a stochastic model of family DD construction based on statistics from the data in conjunction with a single tunable parameter that describes the correlation in DD decision of cells within a family. For each condition, data was pooled across families and the number of cells, $n_k$, observed to have undergone DD in each generation $k$ recorded. From these data, the proportion of cells, $p_k$, that did not

undergo DD in generation $k$ was determined taking cohort correction into account

$$p_k = 1 - \frac{n_k}{\sum_{l \geq k} n_l 2^{(k-l)}}. \qquad (1)$$

If cells within a family made independent DD decisions, a binomial number of them, with probability $p_k$ would be observed to progress without experiencing DD to generation $k+1$. To capture within-family correlated DD fate, the long-established correlated generalization of the binomial distribution, the beta-binomial distribution, was employed. This distribution is parameterized by the probability of progression, $p_k$, and a value $\rho \in [0,1]$ that captures the correlation in the fate of each pair of cells within a family in the same generation. If $\rho = 0$, then all clonal cells make independent DD decisions to progress to the next generation with probability $p_k$. If $\rho = 1$, then all clonal cells in each generation share a single DD decision to progress to the next generation with probability $p_k$. As $\rho$ ranges from 0 to 1, this dependency is interpolated. The proportion of cells across multiple families that progress from one generation to the next is determined by $\{p_k : k = 1, \ldots, 6\}$, irrespective of the value of $\rho$.

Given $\{p_k : k = 1, \ldots, 6\}$, defined by a stimulation condition, and a correlation $\rho$, the induced probability distribution on full pre-sampled DD trees was determined (see Methods). Each cell from a full clonal family is sampled independently with a probability determined either by the per-condition average proportion of beads recovered per well (Fig. 4c) or the average per well volume measured (Supplementary Fig. 6c), from which we computed the sampled-tree range distribution (see Methods).

For a variety of values of $\rho$, including the per-condition maximum likelihood estimate, the resulting distribution of range was determined. With 95% confidence intervals based on the number of families that are experimentally recovered (see Methods), Fig. 4c displays the range distributions that arise from this model overlaid with the experimental data. For all conditions, a substantial correlation of $\rho \geq 0.8$ is necessary to recapitulate the data and, in particular, the range data is not consistent with a family-independent DD mechanism.

**Signal integration effects on clonal division fate**. The above analysis indicates that clonal DD is a strongly inherited,

**Figure 2 | A novel high-throughput clonal assay to measure T-cell division fate. (a)** OT-I/Bcl2l11$^{-/-}$ CD8$^+$ T cells were purified and **(b)** labelled sequentially with different combinations and concentrations of CFSE, CTV and CPD. **(c)** T cells from the 10 different labelling configurations indicated were mixed together and stimulated with N4 peptide $\pm \alpha$CD28 (2 $\mu$g ml$^{-1}$). Between 500 and 2,000 cells from each of all 12 labelling configurations were also cultured separately to use as compensation and gating controls. **(d)** Just prior to first division (26 h) a single cell per labelling configuration from each stimulation condition was sorted into new wells and cultured $\pm$ hIL-2 (1 U ml$^{-1}$). Thus there were four stimulation conditions in total: N4-only, N4 + $\alpha$CD28, N4 + IL-2, N4 + $\alpha$CD28 + IL-2. **(e)** 7,500 beads and propidium iodide (PI) were added per well before analysis to estimate sample recovery and detect dead cells. Cells were carefully transferred to tubes and the complete sample was collected by flow cytometry. Proliferation of clonal progeny cells was measured at 54, 62 and 72 h post stimulation. **(f)** Gates for data analysis were created using control populations at each time-point then applied to the clonal samples. FSC/SSC profile was used to gate beads and lymphocyte populations and then PI exclusion used to identify live cells. Live cells were separated out into differentially labelled populations by classifying cells as CPD$^+$ or CPD$^-$ then plotting CFSE versus CTV to distinguish the division number of cells from different labelling populations. FSC/SSC was then used to classify cells as small, thus having reached their DD (see Supplementary Fig. 2 and Methods for further details on small cell gating).

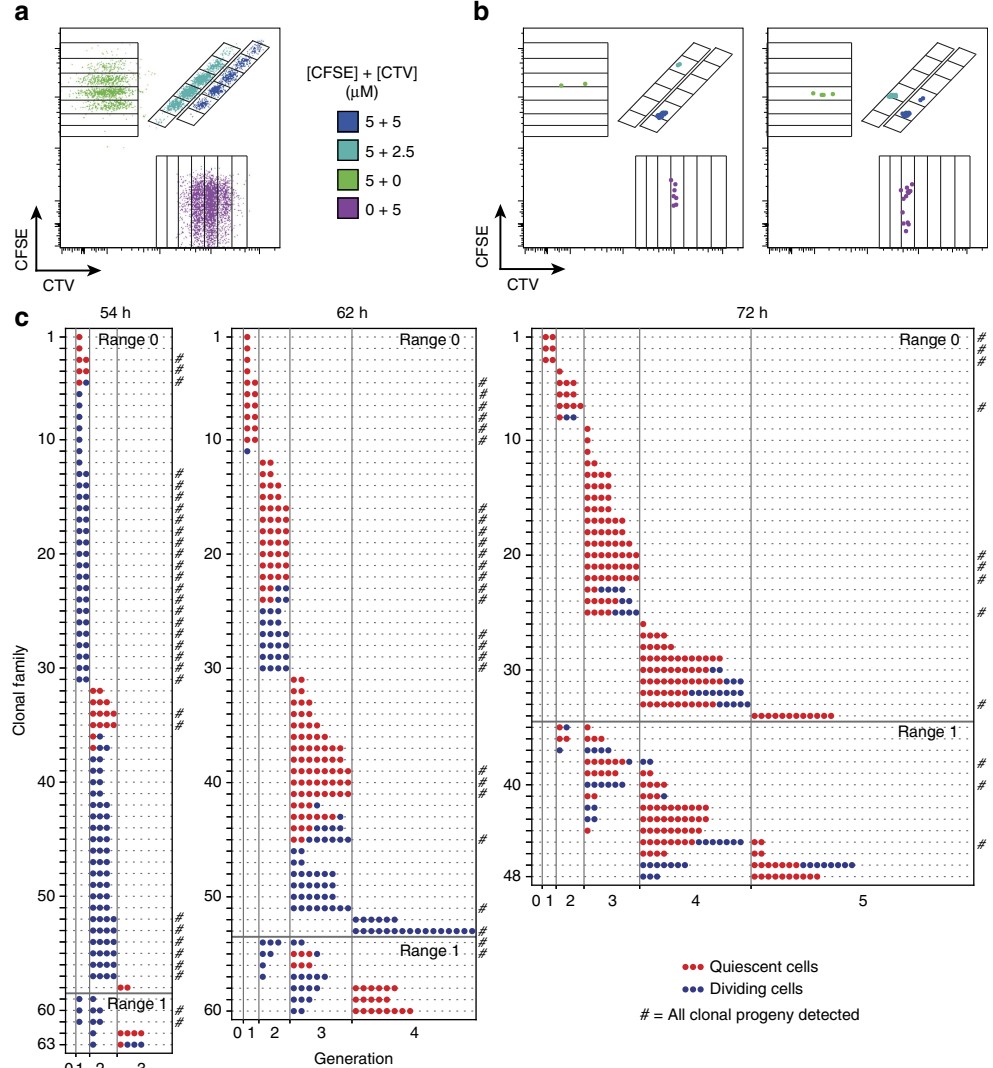

**Figure 3 | Clonal T-cell family proliferation is synchronized.** OT-I/*Bcl2l11*⁻/⁻ CD8⁺ T cells were processed, stimulated and analysed as described in Fig. 2. All cultures contained S4B6 (25 μg ml⁻¹). (**a**) N4 + αCD28 + IL-2 stimulated population of control cells labelled with CTV and CFSE to distinguish generation number for four distinct labelling configurations. Cells were separated into CPD⁺ and CPD⁻, allowing division tracking in eight populations per well. (**b**) Examples of clonal progeny detected in individual wells. Example data shown for 72 h time point. (**c**) Generation number of progeny cells detected from individual clonal families at each time point from N4 + αCD28 + IL-2 stimulation condition. Progeny cells were classified as quiescent based upon small cell size (refer to Methods). Clonal range = maximum − minimum generation number. The symbol # at the end of a line denotes clones where all progeny cells were detected. Founder cell input after sorting was 80, 80 and 64 for data from 54, 62 and 72 h respectively. Results from a second independent experiment are shown in Supplementary Fig. 3.

concordant property within a family lineage under a range of culture conditions. Given this conclusion, we turned our attention to the question of how signal addition at the single cell level operates. Results from signal addition studies at the population level provide a hypothesis[9] as the reported summation had an additive effect on both the mean and variance of the population DD distribution. When the implications of these observations are considered at the family level, this finding suggests that signal integration at the clonal level might be stochastically independent. That is, each signal is responsible for heterogeneous levels of expansion and the T-cell clones independently integrate each signal. To test this prediction, we further interrogated the data presented in Fig. 4.

**Stimuli effects on clonal division fate add independently**. To enable quantitative comparison between the expansion effects of distinct stimuli in combination, we developed a new theory for

the addition of family trees, presented in detail in Methods. The most significant consequence of our mathematical analysis is that in order to determine the generations in which cells become quiescent in a concatenated tree, it is sufficient to know how many cells become quiescent in each generation in the trees to be added (Fig. 1d, Supplementary Fig. 1 and Methods). This feature allows us to address questions of signal integration using data from the multiplex clonal proliferation assay, which does not provide entire family tree structure. Moreover, determination of the generations in which cells are quiescent in the final tree does not depend on how the participating trees are interlaced in the addition procedure (Supplementary Fig. 1 and Methods). This indicates that the mathematics is suitable for describing simultaneous application of mitogenic stimuli, as the DD outcome is invariant to the order of their impact.

For each stimulation condition, we first summarized the DD information of each clonal family with two expansion statistics,

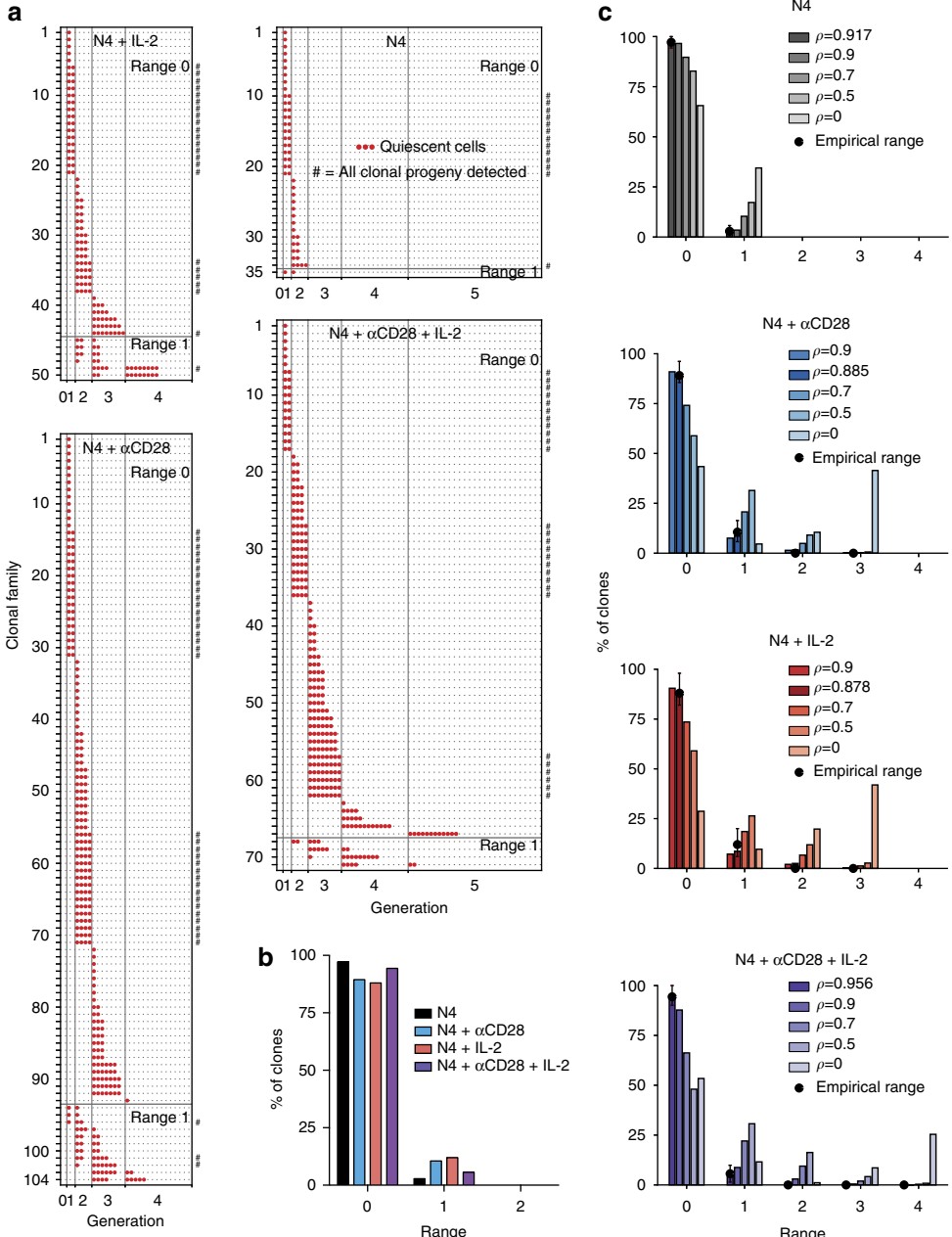

**Figure 4 | Clonal family DD is highly concordant.** OT-I/*Bcl2l11*$^{-/-}$ CD8$^+$ T cells labelled with a division tracking dye multiplex were stimulated with N4 peptide ± αCD28 (2 μg ml$^{-1}$) for 26 h, sorted for one clone per labelling configuration per new well then cultured ± hIL-2 (1 U ml$^{-1}$) as described in Fig. 2. All cultures contained S4B6 (25 μg ml$^{-1}$). (**a**) Generation number in which progeny cells reached DD. Data pooled from 54, 62 and 72 h from families where all detected progeny were quiescent. Founder cell input after sorting was 96, 224, 96 and 224 for N4, N4 + αCD28, N4 + IL-2 and N4 + αCD28 + IL-2, respectively. (**b**) Percentage of clones with concordant (range = 0) or discordant (range > 0) DD. (**c**) To quantitatively question the level of familial correlation in DD required to explain the range data in **b**, a mathematical model was constructed and parameterized by the data and pairwise correlation, ρ, in DD fate (see Methods). The empirical range distribution for each condition is shown (black dots within 95% confidence intervals, see Methods), in addition to model the range distribution for different values of ρ, including the per-condition best-fit. Results from a second independent experiment are shown in Supplementary Fig. 6.

the maximum DD (maxDD), that is, the greatest generation observed in a quiescent family, and the cohort normalized mean DD (mDD) (see Methods). The empirical cumulative distributions of these statistics are plotted in the left panels of Fig. 5a and b, respectively. Crucially, in the framework of tree concatenation both maxDD and mDD are linear operators. Consequently, if signal integration were independent, then the distribution of the maxDD or mDD determined from data generated by two stimuli would necessarily coincide with the convolution of the distributions generated by each stimulus alone. In particular, if the influence of αCD28 and IL-2 on DD were independent, with no more than one of them correlated to the N4 effect, then the distribution of the sum of the maxDD or mDD statistics of (N4 + αCD28) and (N4 + IL-2), which is the convolution of those two distributions, would correspond with the distribution created from the sum of (N4) and (N4 + αCD28 + IL-2). As shown in the right panels of Fig. 5a and b, respectively, these convoluted distributions align

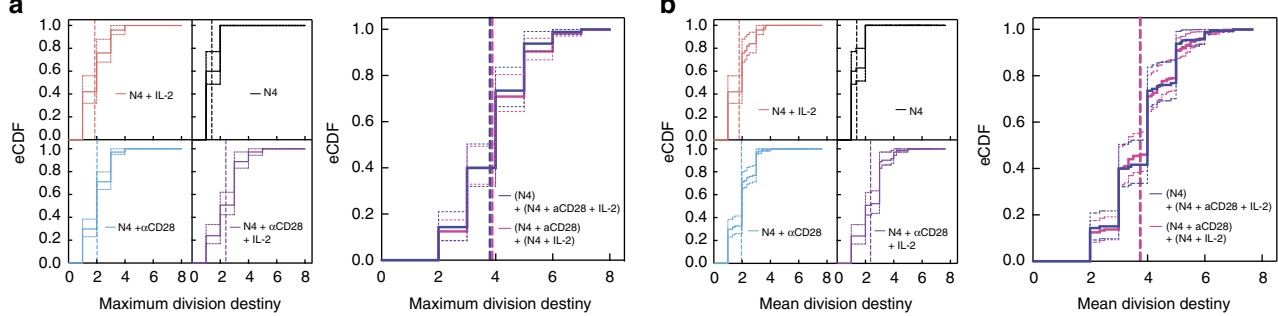

**Figure 5 | Stimuli effects on DD sum independently at the level of the clonal family tree.** OT-I/$Bcl2l11^{-/-}$ CD8$^+$ T cells labelled with a division tracking dye multiplex were stimulated with N4 peptide ± αCD28 (2 µg ml$^{-1}$) for 26 h, sorted for one clone per labelling configuration per new well then cultured ± hIL-2 (1 U ml$^{-1}$) as described in Fig. 2. All cultures contained S4B6 (25 µg ml$^{-1}$). Empirical cumulative distribution functions (eCDF) of measures of clonal expansion (**a**) maximum DD (maxDD) and (**b**) mean DD (mDD) for each individual stimulation condition (left panel). To test clonal signal addition, the convoluted distribution of the statistics from (N4) + (N4 + αCD28 + IL-2) and (N4 + αCD28) + (N4 + IL-2) were compared (right panel, see Methods). Vertical dashed lines represent mean of the pooled clones. Dotted lines show 95% confidence intervals. A non-standard χ$^2$-test of independence (see Methods) was not rejected for either maxDD (P = 0.399) or mDD (P = 0.237). Results from a second independent experiment are shown in Supplementary Fig. 7.

remarkably well (see also, Supplementary Fig. 7). In order to test the statistical hypothesis that the convolutions are independent it was necessary to create a non-standard test based on previous results (explained in Methods), giving P values of 0.399 and 0.237 for maxDD and mDD, respectively, and thus the hypothesis of independent additivity of αCD28 and IL-2 is not rejected.

**Signal sensitivity regulates clonal family DD.** Taken together, the combination of strong clonal concordance in DD and independent additivity of signals traces a large component of DD fate programming to the initial founder cells. Thus we speculated that variation in clonal fates might be traced to measurable differences in levels of cellular components associated with signal perception. In particular, for signal integration to be independent at the clonal level, we reasoned that receptor levels for different signals would be uncorrelated in activated cells. If this were not correct, we expect correlated receptor levels would lead to dependence of signal strength between costimuli, which was not observed.

To identify possible sources of individual founder cell variation we asked whether differences in critical receptor levels for the stimuli tested in Figs 2–5 correlated with subsequent family clone size. CD28 and interleukin-2 receptor chain alpha (IL-2Rα) levels just prior to the first division were relatively uniform (Supplementary Fig. 8) (Spearman's correlation of 0.16), supporting the finding of independence of signal effect on DD in clonal families. To investigate if CD28 levels influenced DD, we sorted naive OT-I/$Bcl2l11^{-/-}$ CD8$^+$ T cells into CD28 high (CD28$^{hi}$) and low (CD28$^{lo}$) expressing populations (Fig. 6a) and measured the effect on mDD. Residual αCD28 antibody from sorting had little effect on cell expansion (open circles, Fig. 6b–d). When αCD28 was added to the culture the CD28$^{hi}$ population had a ~0.6 division increase in mDD relative to CD28$^{lo}$ cells (black arrow, Fig. 6c,d), resulting in an ~50% increase in cell expansion (closed circles, Fig. 6b,c). Thus, differences in initial CD28 receptor level exhibited by the naive T-cell population did contribute to the founder cell variation in DD.

In contrast, sorting for IL-2Rα high (IL-2Rα$^{hi}$) and low (IL-2Rα$^{lo}$) levels prior to first division (Fig. 6e) gave a shift of ~0.73 divisions in mDD irrespective of whether IL-2 was present in the culture (Fig. 6f–h). As IL-2Rα is regulated by the strength of TCR stimulation[11,23,24], the difference in expression between IL-2Rα$^{hi}$ and IL-2Rα$^{lo}$ cells likely reflects intrinsic differences in cellular TCR stimulation strength due to stochastic antigen

encounter, consistent with the observation that TCR stimulation strength regulates DD[9].

The observation that mDD in this system was not affected by IL-2Rα level, but was altered by the ligand IL-2, implies that within the range of IL-2Rα levels found in the stimulated population transmission of signal was not limited by this component of the multi-subunit receptor. We suggest this result is explained by previous studies that demonstrate the IL-2Rα expression is in excess and far exceeds the number of IL-2Rβ and γ chains, the receptor units required to transmit IL-2 signals[25,26]. Given this conclusion, IL-2Rα expression would not affect the independence of IL-2 signal integration with other co-receptors. Thus uncorrelated signal integration is dependent not only on the receptor levels, but the degree to which this variation changes signal sensitivity.

The conclusion that DD fate identifies with IL-2Rα and CD28 expression levels before first division is further illustrated in Supplementary Fig. 9. OT-I/$Bcl2l11^{-/-}$ CD8$^+$ T cells were sorted for CD28$^{hi}$ expression (top 20%), stimulated by N4 self-presentation, then sorted at 26 h for IL-2Rα$^{hi}$ (top 35% of population) and monitored for 72 h in media supplemented with hIL-2. The variation in division fate in these cells was compared with that of an unsorted population control. As predicted, sorting significantly reduced the variation in DD outcomes, with a reduction in the population variance from 1.3 to 0.67.

**Discussion**

Cellular replication is a simple, effective mechanism for generating a large number of antigen-specific clones from rare precursor cells. During an acute response *in vivo* a CD8$^+$ T cell can divide up to 15–20 times[27–29] and give rise to numerous classes of effector and memory cell types, all with identical antigen receptors. *In vivo* single-family tracking studies have revealed significant heterogeneity in the family size of identical cells and a strong concordance in familial differentiation fate[2,3]. Other studies have ascribed variation in fates to early division bifurcations, and numerous extrinsic signals known to influence the pattern of T-cell regulation[14,15,30]. Thus, the relative contribution of extrinsic and intrinsic differences to the fate of otherwise identical cells following stimulation is currently of great interest.

We sought to gain insight into this question by investigating the genesis of heterogeneity following stimulation of apparently identical T cells placed under controlled stimulatory conditions.

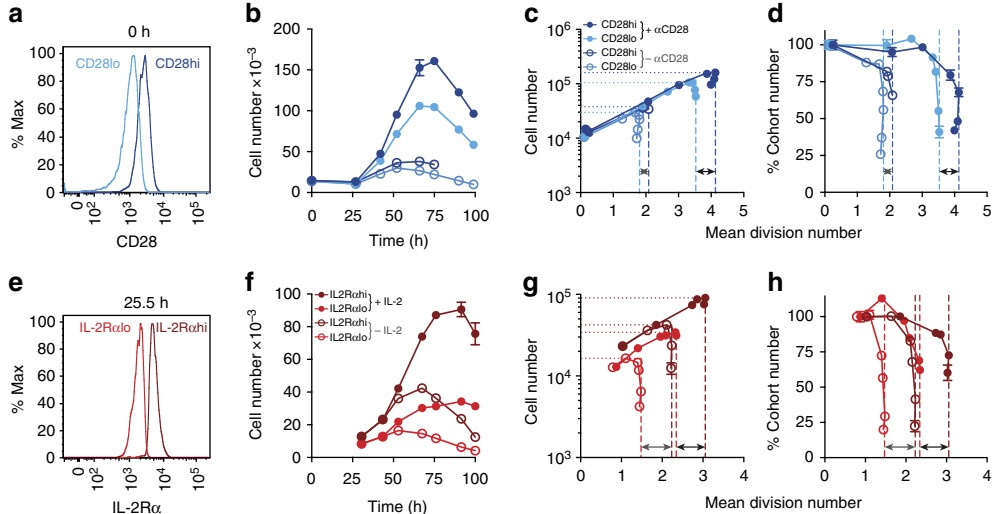

**Figure 6 | Inter-clonal variation in DD is regulated by receptor sensitivity and clonal experience.** (**a–d**) Naive CTV-labelled OT-I/*Bcl2l11*$^{-/-}$ CD8$^+$ T cells were sorted into (**a**) CD28 high and low expressing populations and stimulated with N4 peptide ± αCD28 agonist antibody (20 µg ml$^{-1}$). All cultures contained S4B6 (25 µg ml$^{-1}$). Cell number versus (**b**) time and (**c**) mean division number (MDN) were measured. (**d**) An estimation of the percentage of the starting cells whose progeny are contributing to the response at that time point, calculated by removing the effect of cell expansion (percentage cohort number, see Methods) versus MDN. (**e–h**) Naive CTV-labelled OT-I/*Bcl2l11*$^{-/-}$ CD8$^+$ T cells were stimulated with N4 peptide and αCD28 (2 µg ml$^{-1}$) for 25 h then sorted for (**e**) IL-2Rα high or low expression. Cells were placed back into culture ± hIL-2 (3.16 U ml$^{-1}$) and cell number versus (**f**) time and (**g**) MDN were measured and (**h**) percentage cohort number versus MDN calculated. Arrows indicate the difference in mDD between populations when no additional ligand (grey) or ligand at the specified concentration (black) was added to the culture. Representative of two (**a–d**) or three (**e–h**) independent experiments. Mean ± s.e.m. of triplicate culture wells.

To do so we have introduced a novel multiplex clonal division-tracking assay based on the combinatorial use of division tracking dyes. This method allowed us to detect cell division fate of clonal families of identical TCR transgenic T cells for up to 6–7 generations using the single-cell technology of flow cytometry.

We applied this method to examine the clonal influence on variation in DD, the generation at which a stimulated cell returns to a quiescent state, a feature known to be highly variable within a population of stimulated cells[9,12,13]. The results were unequivocal: our experiments revealed progeny cells from clonal families ceased dividing in the same or adjacent generations with high probability, identifying inter-clonal variation as the principle source of DD diversity under these conditions. This conclusion for T cells aligns with earlier results from B-cell filming that revealed similarly strong concordance in family DD after stimulation by the toll-like receptor agonist CpG DNA[12]. Together these data lead us to a general hypothesis: that stimulation of a resting T or B cell initiates an intrinsic sequence of symmetric divisions with an automated return to quiescence.

Our investigation of DD highlights strong familial concordance raising the question of how branches or discordance might occur. Many studies have shown that T-cell fate is influenced by extrinsic signals[30–32]. The relative contribution of for example, ligands found on an antigen presenting cell (APC) fostering an asymmetry in an early division[14,33], or variation in exposure to APC and different antigen experiences as the cells divide will also percolate through to branching changes in family responses. Depending on where alternative signals are experienced, whole families with alternative branching could be created. Our assay is well suited to investigate the contribution of controlled delivery of extrinsic systems especially where direct filming would be technically difficult, as is the case when stimulation is provided by antigen presenting cells or a stromal cell source is necessary.

For T cells, DD is sensitive to costimulatory and cytokine signals, and our multiplex dye method allowed us to assess the manner of signal addition. To do so we developed a new mathematical framework to study the addition, or concatenation, of family trees, as well as developing a novel statistical test to challenge the hypothesis of independent signal integration. Application of these techniques established that the clonal level effects on generation number of summation of T-cell stimuli are consistent with them being stochastic and independent. Given the simplicity of our system, this conclusion allowed us to trace the source of the stochastic heterogeneity in DD to differences in programming of the initial naive cell.

These findings turned our attention to identifying cellular features and molecular determinants that might account for the differences in DD imprinting in each founder cell. Here the results for CD28 were informative. OT-I/*Bcl2l11*$^{-/-}$ CD8$^+$ T cells express a broad log-normally distributed range of CD28 on the 'identical' population. This variation has significant consequences as the expansion achieved corresponds to the strength of signal integrated and the resulting DD of the cell. The cytokine IL-2 is an important contributor to T-cell responses. Our culture system eliminated autocrine IL-2 contributions that would likely serve to increase the variation in each family tree as local concentrations develop. Furthermore, our analysis of the receptor indicated a broad distribution following this high affinity stimulation level, but receptor numbers were not limiting for the transmission of the IL-2 signal. Published studies show that weaker stimulation leads to lesser expression of the receptor[11,23,24]. For sufficiently weak stimulation, we expect a point will be reached where the IL-2 produced and the receptor level expressed promote variation in DD among similar cells due to changes in local conditions. Examining this possibility requires further *in vitro* studies, followed by *in vivo* experiments to determine its functional significance.

Many additional receptors transmit signals that influence the final extent of expansion and division progression including

CD27, IL12, IL-4, IL-2 and IL-6 (refs 9,10,34,35). Independent stochastic expression of receptor numbers for each of these ligands would result in a multifactorial complex population, where the experience and fate of every cell would be different even when placed in identical stimulation conditions. As the stochastic drivers of receptor heterogeneity are reproducible, we speculate that the inherent variegation of the initial population, and the mix of different fate allocations, will prove to be consistent and robust. Combinatorial differences in DD fate would result from them and, as differentiation to effector states and memory states has been linked to division progression[36,37], this in turn would impact the genesis of different cell types. What, if any, direct influence that these receptor levels have on differentiation programmes warrants further investigation. A combination of the multiplex assay for division tracing described with index sorting would suffice for that purpose.

Our discovery of striking concordance in a controlled *in vitro* setting raises the question, how much of the CD8[+] T-cell response during an *in vivo* infection is driven by an automatous response with multiple concordant family outcomes, and how much is directed by extrinsic influences? Although at this point no direct *in vivo* observations exist to resolve this question we can interrogate the features of existing clonal and population *in vivo* data to test whether it would be consistent with the strongly concordant, inherited DD we have observed *in vitro*. By assuming a family-based DD, our previous *in vivo* population experiments allowed estimation of the generations in which families reached DD[9] from which we determined the relative contribution of clonal families to the response magnitude. Comparing this to findings from published *in vivo* clonal studies[2,3] we found a quantitatively similar pattern, with a small number of clones generating the bulk of the responding cells (Supplementary Fig. 10a). Based on the family sizes from ref. 2 we could also estimate the DD distribution and found the inter-clonal DD was spread over ~10–15 generations, consistent with model fitting to population response data (Supplementary Fig. 10b). Thus, *in vivo* findings are consistent with a clonally regulated DD, though further studies more directly measuring division fate as well as monitoring simultaneous differentiation changes to effector and memory states associated with clonal cell division are required to complete and formally test our understanding of clonal fate regulation under these complex conditions.

Taken together these findings provide fundamental insight into T-cell fate regulation at the level of the individual clone, and we anticipate they will help lead to predictive models of clonal cell fate regulation. Given the growing importance of anti-cancer therapies that expand and reinvigorate highly clonal *in vitro* and *in vivo* T-cell responses[38,39], this better understanding of the fundamental nature and source of the variation in burst-size will facilitate rational optimization of T-cell response manipulation.

## Methods

**Mice.** OT-I/*Bcl2l11*[−/−] and OT-I/FucciRG mice[9] were bred and maintained under specific pathogen-free conditions in the WEHI animal facilities (Parkville, Victoria, Australia) and used between 6–10 weeks of age. OT-I/FucciRG mice were bred from the red (R) FucciG1-#639 and green (G) FucciS/G2/M-#492 mouse lines. All experiments were performed under the approval of the WEHI Animal Ethics Committee.

**CD8[+] T-cell purification.** CD8[+] T cells were isolated from mouse lymph nodes and spleens by negative selection using EasySep Mouse CD8[+] T-cell Isolation kit (StemCell Technologies) according to the manufacturer's protocols. Enrichment of OT-I CD8[+] T cells was confirmed by flow cytometry with a yield of 90–95% CD8[+]Vα2[+] lymphocytes.

**Labelling with division tracking dyes.** OT-I/*Bcl2l11*[−/−] CD8[+] T cells were labelled with the indicated combinations and concentrations of the division

tracking dyes CTV, CFSE (both Invitrogen) and CPD (eBioscience) in PBS (WEHI media) containing 0.1% BSA (Sigma) (PBS 0.1% BSA) at a density of ≤10[7] cells ml[−1] at 37 °C for 20, 10 and 10 min, respectively. The reaction was quenched by washing with 2 × 5 ml ice-cold RPMI 5% FCS.

**In vitro cell culture.** Complete tissue culture medium was RPMI 1640 medium supplemented with 10% FCS, non-essential amino acids, 1 mM Sodium-pyruvate, 10 mM HEPES, 2 mM GlutaMAX, 100 U ml[−1] Penicillin, 100 μg ml[−1] Streptomycin (all Invitrogen) and 50 μM 2β-mercaptoethanol (Sigma). OT-I/*Bcl2l11*[−/−] CD8[+] T cells were stimulated with 0.01 μg ml[−1] SIINFEKL (N4) peptide (Auspep) in 96 well round-bottomed plates by self-presentation at a density of 10,000 cells per well in 200 μl complete tissue culture medium, as described previously[9]. All cultures contained 25 μg ml[−1] of anti-mouse IL-2 monoclonal antibody (supernatant from hybridoma cell line S4B6, WEHI monoclonal antibody facility) that blocks the activity of mouse IL-2 *in vitro* but does not recognize human IL-2 (hIL-2)[9]. Recombinant hIL-2 (Peprotech) and anti-CD28 (clone 37.51, WEHI monoclonal antibody facility) were added to cultures where indicated. Cells were incubated in a humidified environment at 37 °C in 5% CO[2].

**Cell sorting and flow cytometry.** Cell sorting was performed on a FACSAria W or L (BD Biosciences) cell sorter. For IL-2Rα and CD28 level sorting, cells were labelled with anti-CD28-PECy7 (clone 37.51, eBioscience) or anti-CD25-FITC (clone 7D4, BD).

Flow cytometry was performed on a FACSCanto II or LSRFortessa X-20 cytometer (both BD Biosciences). Data were analysed using FlowJo software (Treestar).

A known number of beads (Rainbow calibration particles, BD Biosciences) and propidium iodide (0.2 μg ml[−1], Sigma) was added to samples immediately prior to analysis. The ratio of beads to live cells was used to estimate the absolute cell number. The following monoclonal antibodies were used for the detection of cell surface markers: anti-CD25 -PECy7, or—APC (clone PC61, BD Biosciences) anti-CD28-PECy7 (clone 37.51, eBioscience). Staining was performed in PBS containing 0.1% BSA and 0.1% sodium azide (Sigma). In Supplementary Fig. 8, activated cells were defined as the 50% of cells with the highest FSC fluorescence. Spearman's correlation was calculated using Matlab 2011a's corr function.

**High-throughput clonal multiplex assay to measure DD.** Naive OT-I/*Bcl2l11*[−/−] CD8[+] T cells were purified and sequentially labelled with CFSE (5, 2.5, 0 μM), CTV (5, 2.5, 0 μM) and CDP (5, 0 μM) (Fig. 2a,b). After the population labelling controls were plated, cells from the 10 labelling combinations indicated in Fig. 2c were pooled together and 10,000 cells were added per well for stimulation with N4 peptide in the presence of S4B6, either with or without anti-CD28 (2 μg ml[−1], Fig. 2c). After 26 h (just prior to the first division), cells were sorted so that a single stimulated (estimated based upon high FSC fluorescence) but undivided cell from each fluorescently distinct population was sorted into each 'sample' well of 96-well round-bottomed plates (Fig. 2d). Cells were cultured in the presence of S4B6 either with or without hIL-2 (1 U ml[−1]). Four different stimulation combinations were monitored: N4, N4+αCD28, N4+IL-2 and N4+αCD28+ IL-2. Cells were collected for analysis by flow cytometry at 54, 62 and 72 h post stimulation. At each analysis time point, 7,500 beads were added to measure sample recovery (>90% of the sample for >90% of the tubes in the experiment shown), and PI (0.2 μg ml[−1]) for dead cell exclusion. Samples were carefully transferred into 5 ml polystyrene tubes and entire sample was analysed (Fig. 2e).

For data analysis, gates were set using single label configuration population controls then applied to clonal data as shown in Fig. 2f. Briefly, lymphocytes were identified from FSC/side scatter (SSC) profiles and dead cells excluded using PI. Cells were divided into CPD[−] and CPD[+] then division gating for each labelling configuration was performed on CFSE versus CTV dot plots. Finally cells were gated as 'small' or 'not small' from FSC/SSC profiles to classify cells as quiescent or dividing respectively. Small cell size has previously been demonstrated to be a good surrogate of lymphocyte quiescence[9,12,20]. We further demonstrated this with independent experiments using OT-I/Fucci cell cycle reporter mice, in which cells accumulate the FucciRed reporter when they have reverted to a quiescent state, or express the FucciGreen reporter when progressing through the S/G[2]/M phases of the cell cycle[9,21,40,41]. OT-I/FucciRG CD8[+] T cells were stimulated in similar conditions to those used in the clonal studies and cell size and Fucci reporter fluorescence measured across several time points where the cells were reaching DD to estimate the accuracy of small cell gates to classify cellular quiescence (Supplementary Fig. 2). Forward-scatter side scatter profiles were used to set small cell gates (Supplementary Fig. 2, left columns) then Fucci fluorescence used to gauge the proportion of incorrectly classified cells (that is, FucciG[+] cells that fell within the 'small' gate and FucciR[+] quiescent cells that fell within the 'large' gate). In this example, 3.3%, 2.9% and 4.7% of cells were incorrectly classified by size-based gating at the 50.5, 66 and 73 h time-points, respectively (Supplementary Fig. 2). Extrapolating these error rates to the data shown in Fig. 3 we can estimate that the quiescence status of 165 of the 171 clones in this example has been correctly classified. Collectively, along with previous findings these results indicate

that small cell size is an accurate method to estimate cellular quiescence in these studies.

**Population DD measurements.** The definition and methods by which DD can be estimated during a population response have been published previously[9]. Briefly, the cohort number (an estimate of the number of starting cells whose progeny are contributing to the response at a time point) was calculated by dividing the cell number per division by $2^i$, where $i$ is the cell's generation. The population mean division number was calculated as the arithmetic mean of the cohort numbers at each time point. Assuming little death, the mean division number will increase in time, plateauing at the point where the cells reach DD. Thus, the population mean division destiny (mDD) was estimated as the maximum mean division number measured over all the time points.

**Clonal DD calculations.** The maximum and minimum generation numbers for a clone were defined as the largest and smallest generation detected within a clone. The clonal range was calculated as the maximum less minimum generation number. T-cell clones were defined as having reached DD based on small cell size[9,12,20] gated from FSC/SSC profiles of control T-cell populations (Fig. 2f). To determine the mDD for individual clones, the data were cohort-normalized as described for population data. With initial clones being in generation 0, $n_i$ being the number of cells in the clone that experience DD in generation $i$ and $C_i = n_i 2^{-i}$ being the cohort-normalized number of progeny cells in generation $i$, the proportion of the clone reaching DD (that is, becoming quiescent (qui)) in each division was normalized to exclude cells lost due to incomplete recovery using the following equation:

$$p_i^{\text{qui}} = \frac{C_i}{\sum_{i=0}^{I} C_i}. \tag{2}$$

Note that after renormalization $\sum_{i=0}^{I} p_i^{\text{qui}} = 1$ for each clone. The mDD for a clone was calculated as follows:

$$\text{mDD} = \sum_{i=0}^{I} i \, p_i^{\text{qui}}. \tag{3}$$

The maximum DD (maxDD) for a clone was defined to be the largest generation observed within a clone in which all the cells had reached DD. Clones were analysed for DD properties in Figs 4 and 5 and Supplementary Figs 5–7 if all progeny cells were considered to have reached DD based on small cell size.

**The beta-binomial model and clonal range data.** For each of the four stimulation conditions, the following computations were performed to determine the clonal range distributions reported in Fig. 4c and Supplementary Fig. 6c. As described in the main text, the conditional probability that a cell progresses to a DD past generation $k$, given it got to generation $k$, $p_k$, was determined from the data. All potential full trees, that is, trees for which all cells are fully sampled, of a given depth were identified by brute force enumeration. Letting $t_i$ be the number of DD cells in generation $i$ for such a full tree, $\mathbf{T}$, the number of cells that have not yet reached DD by generation $i$ can be determined as $r_i = (1 - \sum_{j<i} t_j 2^{-j}) 2^i$. With $B(n,k|p,\rho)$ denoting the beta-binomial distribution for picking $k$ objects out of $n$, parameterized by probability $p$ and correlation $\rho$, the likelihood of the full tree is $P(\mathbf{T} \mid p_1, \ldots, p_K, \rho) = \prod_{i=1,\ldots,K} B(r_i, t_i \mid p_i, \rho)$. For any sub-tree, $\mathbf{S}$ of $\mathbf{T}$, that is, such that $s_i \leq t_i$ for all $i$, the likelihood of $\mathbf{S}$ given the sampling probability $r$ is $P(\mathbf{S} \mid \mathbf{T}, r) = \prod_{i=1,\ldots,K} B(t_i, s_i \mid r, 0)$. Finally, the probability of observing range $f$ is

$$P(f \mid r, p_1, \ldots, p_K, \rho) =$$
$$\sum_{\mathbf{T}} \sum_{\mathbf{S} \leq \mathbf{T}} \chi(\text{Range}(\mathbf{S}) = f) P(\mathbf{S}|\mathbf{T},r) P(\mathbf{T}|p_1, \ldots, p_K, \rho), \tag{4}$$

where $\chi$ is the indicator function and the sum is over all full trees $\mathbf{T}$ and all sub-trees $\mathbf{S}$ of the full tree $\mathbf{T}$, and Range($\mathbf{S}$) is the range of the sub-tree $\mathbf{S}$. The per-stimulation condition best-fit values of $\rho$ displayed in Fig. 4c and Supplementary Fig. 6c were the maximum likelihood estimates identified by custom Python and MATLAB code.

**A mathematical framework for the addition of family trees.** To investigate whether the effect of stimulatory signals were independently integrated by families, it was necessary to describe the addition or concatenation of family trees. If signals are administered sequentially, a natural definition is to append the family tree generated by the second signal to the DD cells of the family tree generated by the first. If the DD of all cells within the tree generated by the first signal is $g_1$ and the DD of all cells in the tree generated by the second is $g_2$, then the sum of two trees will result in one where every cell has a DD of $g_1 + g_2$ (Fig. 1d, top panel) irrespective of the order of signal administration. If there is variation of DD of cells within each family tree, however, then even with sequential signal administration the resulting tree will differ depending on the order of administration (Fig. 1d, bottom panel).

Despite the non-commutativity of the tree shape, the generation of quiescent cells in the tree created by the sum does not depend on the particulars of the addition and can be determined from knowledge of the DD of cells in each tree without reference to the tree structure. This desirable property extends to the circumstances of non-commutativity of administration or action of signals. If two or more mitogenic signals are added to culture simultaneously, there are many possible tree additions depending on the time and place in the clonal tree these signals provide their contribution. An example of the possible clonal outcomes from different permutations of interlacing signal addition is illustrated in Supplementary Fig. 1. The fundamental premise of this definition of additivity is that the cells in the summed tree experience the expansion aspect of each signal once.

That the generation of quiescent cells in the summed tree is invariant to the particulars of the tree addition is established by the following result. If $\mathbf{x}$ and $\mathbf{y}$ are two vectors whose $i$th components, $x_i$ and $y_i$, report the number of cells in that family tree whose DD is in generation $i$, then the equivalent vector for the summed tree, irrespective of how the interlacing of trees is done, is given by the convolution of the vectors $\mathbf{x}$ and $\mathbf{y}$, $\mathbf{v} = \mathbf{x} * \mathbf{y}$, where $v_i = \sum_{j+k=i} x_j y_k$.

To see this, note we have a correspondence between a family tree and a directed tree by identifying the progenitor with the root vertex and the quiescent cells with the leaves of the graph. For any directed tree, we can associate to it a generating function for its leaves[42]:

$$X(s) = x_0 + x_1 s + x_2 s^2 + x_3 s^3 + \ldots \tag{5}$$

where $x_i$ is the number of quiescent cells in generation $i$ and $s$ is a real number.

Given two trees with generating functions $X(s)$ and $Y(s)$, the generating function of the tree created by appending a copy of the second tree to each quiescent cell of the first tree is $X(s)Y(s)$. This can be seen by observing that for each quiescent cell in the first tree of generation $i$, in the appended tree we now have a contribution to the generating function of $s^i(y_0 + y_1 s + y_2 s^2 + y_3 s^3 + \ldots)$, as every quiescent cell in the first tree gives rise to quiescent cells of the second tree, but with $i$ added to their generation. While appending copies of the second tree to the quiescent cells of the first may not result in a tree that is isomorphic to the tree created by appending copies of the first tree to the second tree, they both have generating functions $X(s)Y(s)$. The $i$th coefficient of this polynomial corresponds to the number of quiescent cells in generation $i$ in the new tree and algebra shows that it is $\sum_{j+k=i} x_j y_k$, as stated above.

Having established the result for appending copies of one tree to the final cells of the other, the next step is the situation where the whole second tree is inserted at a final-cell partition of vertices of the first tree. Let $\{n_j : j = 1,\ldots,J\}$ be a collection of vertices in the first tree whose descendants form a partition of all of the final cells of the original tree. If $g(j)$ denotes the generation of the vertex $n_j$ and $X_j(s)$ denotes the generating function of the number of quiescent cells in each generation of the sub-tree rooted at $n_j$, due to the descendent partition requirement, we have

$$X(s) = \sum_{j=1,\ldots,J} s^{g(j)} X_j(s). \tag{6}$$

Inserting the second tree at each vertex $n_j$ and end-appending copies of the remaining first tree results in a generating function $Y(s)X_j(s)$ by the earlier result. Thus the overall generating function is again

$$\sum_{j=1,\ldots,J} s^{g(j)} Y(s) X_j(s) = Y(s) \left( \sum_{j=1,\ldots,J} s^{g(j)} X_j(s) \right) = X(s)Y(s) \tag{7}$$

and the number of cells in DD in each generation is the same as before. Extending the result to more general interlacing follows arguments along the same lines recursively. Thus, no matter how the additive signal integration occurs, the DD of the cells of an added tree can be determined by knowing the DD of the cells in the trees to be summed, which is the information available from the multiplex method.

**Linearity of mDD and maxDD.** Within this framework of tree addition, the summary statistics mDD and maxDD can be shown to all be linear functions. Consider two trees described by vectors $\mathbf{x}$ and $\mathbf{y}$. For maxDD of the summed tree, we have

$$\text{maxDD}(\mathbf{x} * \mathbf{y}) = \max\left( i : \sum_{j+k=i} x_j y_k > 0 \right) \tag{8}$$

and this maximum is attained when $i = \text{maxDD}(\mathbf{x}) + \text{maxDD}(\mathbf{y})$. Linearity of mDD follows directly from an expansion of terms.

**Testing clonal level independence of signal integration.** Figure 4a shows that each stimulation environment results in heterogeneous family trees and so must be regarded as stochastic. As the statistics maxDD and mDD are linear, if primary and costimulatory signal integration was independent at the level of clones, then the distribution of those statistics for a combination of signals would be the convolution of the statistics taken from individual costimulatory environments. It is not possible to directly measure the expansion effect of these costimulatory signals in isolation, as they require primary stimulation, achieved in this case with N4. Regardless, under the hypothesis of independent signal integration, the distribution of maxDD(N4 + αCD28) + maxDD(N4 + hIL-2) should equal that of

maxDD(N4) + maxDD(N4 + αCD28 + hIL-2). Along with the same result using mDD, that is what we compare.

**Estimating the CDF of a sum of independent random variables.** The maximum likelihood estimator (MLE) for the cumulative distribution function (CDF)[43] of the sum of two independent random variables, given observations of each of them, is the convolution of their empirical CDFs (eCDFs). That is, given independent and identically distributed observations of $X_1, X_2, \ldots, X_n$ and $Y_1, Y_2, \ldots, Y_m$, we have

$$
\begin{aligned}
&\max P(x_1, \ldots, x_n, y_1, \ldots, y_m \mid F^{X+Y}) \\
&= \max P(x_1, \ldots, x_n, y_1, \ldots, y_m \mid F^X, F^Y s.t. F^{X+Y} = F^X * F^Y) \\
&= \max P(x_1, \ldots, x_n \mid F^X) \max P(y_1, \ldots, y_m \mid F^Y).
\end{aligned}
\tag{9}
$$

Thus the MLE for the CDF of $X + Y$ given $n$ observations of $X$ and $m$ observations of $Y$ is the convolution of the eCDF of $X$ with the eCDF of $Y$:

$$
F_{n,m}^{X+Y}(z) = (F_n^X * F_m^Y)(z) = \frac{1}{nm} \sum_{i=1}^{n} \sum_{j=1}^{m} \chi(x_i + y_j \le z)
\tag{10}
$$

where $\chi(x_i + y_j \le z)$ is 1 if $x_i + y_j \le z$ and 0 otherwise.

To determine the eCDF of the mDD and maxDD of a sum of two conditions assuming that the DD influence of the signals was independent, we evaluated the eCDF for each of the signals and convolved those using custom software executed in Matlab and Python to obtain the MLE CDF for the summed condition. This procedure led to Fig. 5 and Supplementary Fig. 7.

**Bootstrap confidence intervals.** The reported asymmetric 95% confidence Intervals (CIs) were generated by creating bootstrap samples of each individual condition, determining their eCDFs[44], convolving those bootstrap eCDFs across two conditions to obtain an estimate of the CDF of the impact of the summed signals, and using percentile bootstrap. Given observations $x_1, \ldots, x_n$ and of $y_1, \ldots, y_m$ of, say, maxDD of two conditions, we created $K = 10{,}000$ bootstrap samples of each data set and determined the MLE of the CDF of their sum $F_{n,m}^{X+Y,k} = F_n^{X,k} * F_m^{Y,k}$ for $k = 1, \ldots, K$. To determine the lower, $L(z)$, and upper, $U(z)$, CIs at each point $z$ of the MLE of the CDF we solve the following minimization problem

$$
(L(z), U(z)) = \operatorname*{argmin}_{l,u} \left\{
\begin{array}{l}
\frac{1}{K}\left(\sum_{k=1}^{K} \chi(-l \le F_{n,m}^{X+Y}(z) - F_{n,m}^{X+Y,k}(z) \le u)\right): \\
0.95 \le \frac{1}{K}\left(\sum_{k=1}^{K} \chi(-l \le F_{n,m}^{X+Y}(z) - F_{n,m}^{X+Y,k}(z) \le u)\right)
\end{array}
\right\}.
\tag{11}
$$

**Test for equality of convoluted distributions.** Pearson's $\chi^2$-test of independence is a classic tool that assesses whether one can reject the hypothesis that two discrete random variables $X$ and $A$, with probability vectors $\mathbf{x}$ and $\mathbf{a}$, respectively, are equally distributed. Assessing whether $X + Y$ is equally distributed as $A + B$ based on $n_X, n_Y, n_A, n_B$, observation of four discrete independent random variables $X, Y, A, B$ with probability vectors $\mathbf{x}, \mathbf{y}, \mathbf{a}, \mathbf{b}$, the hypothesis to be tested becomes

$$
\mathbf{H_0} : \mathbf{x} * \mathbf{y} = \mathbf{a} * \mathbf{b}
\tag{12}
$$

where $*$ denotes convolution. As the covariance structure of $\mathbf{x} * \mathbf{y}$ is no longer that of a multinomial, Pearson's statistic is no longer $\chi^2$ distributed and another test needs to be developed. We achieve this via Theorem 2 of ref. 45.

Under $\mathbf{H_0}$, it can be shown that the sequence $\mathbf{V} = n_X^{-1/2}(\mathbf{X} * \mathbf{Y}/(n_X n_Y) - \mathbf{A} * \mathbf{B}/(n_A n_B))$, created from empirical observations, is centred and normally distributed with a covariance matrix $\Sigma$, determined below, that is distinct from the covariance matrix in the usual multinomial test of independence. Using a consistent estimator of a Moore–Penrose pseudo inverse $\Sigma^{+,\mathbf{e}}$ of $\Sigma$, the statistic

$$
S = \mathbf{V} \Sigma^{+,\mathbf{e}} \mathbf{V}'
\tag{13}
$$

is $\chi^2$ distributed with $\operatorname{rank}(\Sigma) = \operatorname{trace}(\Sigma^+ \Sigma)$ degrees of freedom. The test's $P$ value is

$$
P(\chi^2(\operatorname{trace}(\Sigma^+ \Sigma)) \ge S).
\tag{14}
$$

To establish these results, we used the central limit theorem[46] and independence of $\mathbf{X}$ and $\mathbf{Y}$ to verify that

$$
n_X^{-1/2}((\mathbf{X}/n_X, \mathbf{Y}/n_Y) - (\mathbf{x}, \mathbf{y}))
\tag{15}
$$

is asymptotically centred and normally distributed with covariance matrix $\Sigma_{(X,Y)} = \operatorname{diag}(\Sigma_X, c_Y^{-1} \Sigma_Y)$, a block diagonal matrix where $c_Y = n_Y/n_X$. Applying the delta method[46] to $(\mathbf{X}/n_X, \mathbf{Y}/n_Y)$ with the convolution operation,

$$
n_X^{-1/2}(\mathbf{X} * \mathbf{Y}/(n_X n_Y) - \mathbf{x} * \mathbf{y})
\tag{16}
$$

is asymptotically centred and normally distributed with covariance matrix $T_Y \Sigma_X T_Y' + c_Y^{-1} T_X \Sigma_Y T_X'$ where $(T_X)_{i,j} = \mathrm{d}(\mathbf{u} * \mathbf{v})_i / \mathrm{d}\mathbf{u}_j \mid_{(\mathbf{u},\mathbf{v})=(\mathbf{x},\mathbf{y})}$ and

$(T_Y)_{i,j} = \mathrm{d}(\mathbf{u} * \mathbf{v})_i / \mathrm{d}\mathbf{v}_j \mid_{(\mathbf{u},\mathbf{v})=(\mathbf{x},\mathbf{y})}$. Similarly,

$$
n_X^{-1/2}(\mathbf{A} * \mathbf{B}/(n_A n_B) - \mathbf{a} * \mathbf{b})
\tag{17}
$$

is asymptotically centred and normally distributed with covariance matrix $c_A^{-1} T_B \Sigma_A T_B' + c_B^{-1} T_A \Sigma_B T_A'$, where $c_A = n_A/n_X$ and $c_B = n_B/n_X$. We apply once more the delta method to the joint vector $(\mathbf{X} * \mathbf{Y}/(n_X n_Y) - \mathbf{x} * \mathbf{y}, \mathbf{A} * \mathbf{B}/(n_A n_B) - \mathbf{a} * \mathbf{b}$, taking their difference, giving

$$
\mathbf{V} = n_X^{-1/2}(\mathbf{X} * \mathbf{Y}/(n_X n_Y) - \mathbf{A} * \mathbf{B}/(n_A n_B))
\tag{18}
$$

is asymptotically centred and normally distributed with covariance matrix $T_Y \Sigma_X T_Y' + c_Y^{-1} T_X \Sigma_Y T_X' + c_A^{-1} T_B \Sigma_A T_B' + c_B^{-1} T_A \Sigma_B T_A'$. Thus the covariance matrix described above has the form $\Sigma = T_Y \Sigma_X T_Y' + c_Y^{-1} T_X \Sigma_Y T_X' + c_A^{-1} T_B \Sigma_A T_B' + c_B^{-1} T_A \Sigma_B T_A'$. As consequence of Theorem 2 of ref. 45, we have that $S = \mathbf{V} \Sigma^{+,\mathbf{e}} \mathbf{V}'$ is $\chi^2$ distributed with $\operatorname{rank}(\Sigma)$ degrees of freedom, where $\Sigma^{+,\mathbf{e}}$ is a consistent estimator of the pseudo inverse $\Sigma^+$ of $\Sigma$, which, assuming consistent ranks, can be obtained by substituting $\mathbf{x}, \mathbf{y}, \mathbf{a}, \mathbf{b}$ with their consistent estimators $n_X^{-1} \mathbf{X}, n_Y^{-1} \mathbf{Y}, n_A^{-1} \mathbf{A}, n_B^{-1} \mathbf{B}$ into $\Sigma$ and evaluating a pseudo inverse of it.

As with all categorical hypothesis tests, for practical use each category needs to have a sufficient number of observations to ensure accuracy of the asymptotic normality approximation and categories need to be combined if the data are too sparse. For the convolution test, this is achieved post convolution by projecting each convolution to a common categorization and then taking the vector difference. As projection is a linear operator, under the null hypothesis, the resulting difference is an asymptotically centred normal distribution whose covariance can be determined. Hence a $\chi^2$ statistic can be created by use of the covariance's pseudo-inverse as above.

For maxDD, no combining of categories was necessary. For mDD, data lying between two consecutive integers, $(i, i+1)$, were binned into a single category. Thus the 11 bins used for mDD were $\{2\}$, $(2,3)$, $\{3\}$, $(3,4)$, $\{4\}$, $(4,5)$, $\{5\}$, $(5,6)$, $\{6\}$, $(6,7)[7,+\infty)$.

**Estimating clonal contributions to _in vivo_ population DD.** To calculate the percentage contribution of clonal families to the magnitude of the total response (Supplementary Fig. 10), it was assumed that all progeny cells would adopt a concordant DD. The probability of a clone reaching DD in a given division was obtained from Cyton fitting to the OT-I/FucciRG CD8$^+$ T cell _in vivo_ influenza (HKx31-OVA) infection data in Fig. 1d,e and Supplementary Table 1 of ref. 9. The discretized probability function was multiplied by the mean initial cell number for the two experiments ($mN_0 = 1{,}808$, Supplementary Table 1 of ref. 9) and only divisions that contained at least one clone were used to determine clonal family contribution to response magnitude (that is, divisions 4–19). The probabilities in each division were normalized so that the discretized probability distribution ($f_i$) for $i \in [4,19]$ summed to 1. The number of progeny cells produced by clones reaching DD in division $i$ was corrected to reintroduce the effects of cell expansion as follows:

$$
N_i^{\mathrm{qui}} = mN_0 \times f_i \times 2^i.
\tag{19}
$$

The percentage contribution to the total response magnitude of progeny cells reaching DD in each division was calculated as follows:

$$
\% N_i^{\mathrm{qui}} = \frac{N_i^{\mathrm{qui}}}{\sum_{i=4}^{i=19} N_i^{\mathrm{qui}}} \times 100.
\tag{20}
$$

This was then plotted as a cumulative function against the percentage of the total clones that generated these cells.

**Inference of DD distribution from _in vivo_ clonal studies.** The percentage contribution of clonal families to the magnitude of the total response was obtained directly from ref. 2. To estimate the DD distribution for this _in vivo_ clonal data, we assumed that _in vivo_ DD was concordant and that minimal cell death had occurred at the time point measured. We estimated the DD as $\log_2(N)$, where $N$ is the total number of progeny cells detected per clone. Clonal DD was rounded up to the next integer value and binned for every second division.

**Code availability.** The computer codes that are used in this study are available from the corresponding authors on request.

**Data availability.** The data that support the findings of this study are available from the corresponding author upon request.

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

## Acknowledgements

We thank the Busch and Schumacher laboratories for providing raw data from their published work for additional analyses in Supplementary Fig. 10, Philippe Bouillet for *Bcl2l11*$^{-/-}$ mice and Manuela Hancock for technical assistance. Fucci Red and Fucci Green mice were provided by the Riken BioResource Centre through the National BioResource Project of the Ministry of Education, Culture, Sports, Science, and Technology of Japan. This work was supported by the National Health and Medical Research Council of Australia via Project Grant 1010654, Program Grant 1054925 and a fellowship to P.D.H. as well as an Australian Government National Health and Medical Research Council Independent Research Institutes Infrastructure Support Scheme Grant 361646. K.R.D. was supported by Science Foundation Ireland Grant 12IP1263. The research leading to these results has received funding from the European Union Seventh Framework Programme (FP7/2007–2013) under grant agreement 317040 (QuanTI). We also received support from the Victorian State Government Operational Infra-structure scheme. J.M.M. was the recipient of an Australian Postgraduate Award, an Edith Moffat Scholarship from The Walter and Eliza Hall Institute and Sydney Parker Smith Postdoctoral Research Fellowship from the Cancer Council of Victoria.

## Author contributions

J.M.M. performed experiments, analysed and interpreted data and co-wrote manuscript. K.R.D. and P.D.H. co-designed the multiplex assay, oversaw study planning, analysis, data interpretation and writing the manuscript. G.P. and K.R.D. developed and performed novel statistical analyses on clonal data. A.K. contributed to data analysis and interpretation. S.H. contributed to experimental planning and data interpretation.

## Additional information

**Competing financial interests:** The authors declare no competing financial interests.

**How to cite this article**: Marchingo, J. M. *et al.* T-cell stimuli independently sum to regulate an inherited clonal division fate . *Nat. Commun.* **7**, 13540 doi: 10.1038/ncomms13540 (2016).

**Publisher's note**: 

