## [Peer Review File · Nature Communications]

Reviewers' comments:

Reviewer #1 (Remarks to the Author):

In their paper "T cell stimuli independently sum to regulate an inherited clonal division fate," Marchingo et al study the nature and source of variation in clonal expansion of stimulated T cells. The main question they asked is why a population of (essentially) identical T cells displays heterogeneous division destiny(DD) (fig 1a)?

The authors proposed two scenarios that could lead to this phenomenon: 1) Clonal concordance (fig 1b top). Cells from the same clonal family have the same DD, but different clonal families may have different DD's. This implies that DD is lineage imprinted. 2) Clonal discordance (fig 1b bottom). Cells from the same clonal family have different DD's. In this case, DD could be either pre-programmed (e.g. asymmetric division) or stochastically regulated. The authors then developed an interesting multiplex-dye assay. The authors claim, based on their data (fig 2c) and model, that scenario 1 is favored.

At this point, the authors abruptly changed to another question: whether the signal integration at the clonal level was stochastically independent. Here, we do not see the connection of this question to their main question of clonal concordance. In any case, the authors developed a mathematical framework and statistical test to support the claim that signal integration is independent. An interesting inference from this independence is "In particular, for signal integration to be independent at the clonal level, it appears necessary that receptor levels for different signals be uncorrelated in activated cells." However, the authors did not dig deeper into this direction.

Overall, there are some interesting ideas, assay and analytical framework are presented. However, we do not see a strong connection between the points they make based on the writing. Also, the authors did not dig deeply into their two claims: clonal concordance and independent signal integration. Finally, multiple assumptions in the model were not supported by data or discussion, which makes interpreting their results challenging. For these major claims, only one experiment is not sufficient. At this time, the study seems premature for publication.

Below we provide some feedback on points of concern/confusion that we hope the authors will find constructive.

1. How can we know whether the observed clonal concordance is not simply due to the small number of divisions allowed before measurements were taken? A simulation of inherited vs. stochastically regulated division destinies would help determine how likely it is that the clonal concordance observed is truly due to an inherited property, as opposed to an effect of taking measurements after only a few divisions in which the cells are able to diverge. If the downstream analysis were limited to clones where at least half of the progeny were measured and all measured cells were quiescent, would the conclusion still hold?
2. Independent experiments are not combined in main figures nor are they given in the supplement. Only one experiment is performed per main claim, and alternative hypotheses are neither mentioned nor refuted.
3. We did not understand the connection between the investigation of clonal concordance and the independent additivity of signals.
4. Controls for quiescence: It is mentioned briefly that the gates chosen for quiescence were based on control cells, but this data is not shown. An accurate measure of quiescence is essential to the arguments made in the paper. Some experiments to demonstrate these cells are indeed quiescent may include inducing quiescence in the derived T cells to confirm that this size gating accurately segregates quiescent cells, or sorting out the cells within this gate and demonstrating that they have ceased dividing. The authors' previous work used FUCCI cells to determine gates for quiescence, but the mice in this manuscript do not appear to have FUCCI.

5. Why was 2.5 CFSE + 5 CTV removed from the analysis? This omission isn't clearly explained.
6. Intra-clone experiments are not repeatable, and cannot be assessed over time: Because of the nature of the method, there is no way to repeat experiments from the same clone. How do the authors conclusively distinguish between hypotheses such as genetic heterogeneity acquired over the lifetime of the mouse vs. the stochastic effects of recent T-cell stimuli?
7. Figure 3: Since the authors are comparing two distributions, why wasn't a KS or rank-sum test used, instead of making asymptotic assumptions in the modified chi-squared test?
8. Variability of the initial labelling: In Figure 2, cells at t26 were selected based on "higher FSC fluorescence" and "being undivided" when sorted. Please clarify whether these cells were from the highest level of fluorescence observed (undivided). An experiment where cells are stained and immediately analyzed would establish the undivided fluorescence level. Additionally, a tight distribution of staining at this time would confirm that the stain intensity accurately reflects the number of divisions, and that variability in staining does not affect conclusions.

Reviewer #2 (Remarks to the Author):

Comments for the authors:

In the manuscript entitled "T cell stimuli independently sum to regulate an inherited clonal division fate" the authors present a very elegant and stringently controlled assay for simultaneously tracking up to six cell divisions of multiple T cell clones harboring identical TCRs in vitro. Using this assay, they aim to decipher when and how the highly distinct clonal burst sizes, recently discovered to originate from single T cells responding to infection in vivo, are determined. The central question they pose is "whether clonal outcomes are intrinsically inherited from the initial cell or if they arise from stochastic extrinsic regulation" of division activity in the initial cell's progeny.

From their data the authors draw the following conclusions:

- 1) The division fate of a T cell clone is imprinted onto the initial cell and intrinsically inherited by its daughters.
- 2) The relevant T cell stimuli received by the initial cell vary stochastically and independently from one another.
- 3) Variation in clonal fate results from stochastic variation of costimulatory signal receptivity (before T cell activation) and variation in T cell receptor signal strength (during T cell activation).

While appreciating the stringency of the well-controlled in vitro system provided by Duffy et al. this reviewer has some substantial concerns as to the validity of the far-reaching conclusions drawn from this dataset.

Main concerns:

- 1) The authors in vitro system (utilizing self-presentation, murine-IL-2 blocking antibody and addition of human-IL-2) deliberately limits the availability of extrinsic signals acting upon the individual members of an expanding T cell clone and thus, does not allow the conclusion that that these factors are generally irrelevant.

As stated by the same group in Marchingo et al. Science 2014: "The strong effect of autocrine interleukin-2 (IL-2) was controlled by adding blocking antibody (clone S4B6) and using human IL-2 (hIL-2), resistant to S4B6, when required." Further experiments are required to resolve whether physiological availability of autocrine IL-2 increases intra-clonal variation.

Further on, the use of self-presentation limits the availability of diverse signals provided by professional APCs. In Extended Data Fig. 5 the authors show that prolonged presence of self-presenting cells in cultures does not lead to substantial intra-clonal variation. However, this could be due to the fact that these non-professional antigen presenters do not provide the signals

relevant for intra-clonal variation. The remaining question is, whether priming via professional APCs and/or their prolonged presence in culture will increase intra-clonal variation?

2) The authors' conclusion "that the summation of T-cell stimuli effects on division number at the clonal level are stochastic and independent" could be substantially confounded by the specific setup of their experiments.

In the authors' in vitro system, TCR stimuli are provided by self-presentation and co-stimulation is provided by soluble anti-CD28 antibody. Thereby, the authors' artificially detach factors from one another that are physiologically provided together - in the cellular context of the APC.

Interdependence of TCR stimuli and CD28 signaling could well arise, if both signals were simultaneously provided by an APC. The physiological situation could look like this: An APC stochastically loaded with more peptide (higher pMHCI density) than another, will interact longer with a given T cell and thus provide more CD28-signaling via CD80/CD86 to this T cell (potentially overriding the effects of subtle variations of CD28 expression in naïve T cells). It thereby remains unresolved whether the use of professional APCs will abrogate the independent manner in which stimuli related to the TCR and to CD28 influence division fate?

3) The authors' conclusion that "clonal heterogeneity arises through differences in stochastic antigen interaction and initial receptor sensitivity" is only partially supported by their data. In support of their conclusion the authors show that sorted CD28-high/low and IL2Ra-high/low subpopulations will display distinct mean division fates. However, they do not investigate the impact of this intervention on variation of clonal division fate directly. Thus, they do not answer how much of the variability found in their system can actually be attributed to the suggested factors - and how much is left unexplained. In the opinion of this reviewer, the impact of reducing CD28 and IL2Ra variability, should best be investigated in a setting in which strong inter-clonal variability is detected to start out with (e.g. Fig. 2C / 72h + hIL2 + anti-CD28).

The question would be, how much the variation of division fate between single clones can be homogenized if clones are preselected for defined criteria? This reviewer would suggest the following experiment: A Population of naïve OTI cells is sorted for CD28^{hi} expression, then stimulated by self-presentation, then single cell-sorted at 26h for IL2R^{hi} and then monitored for 72h in media supplemented with hIL2.

Minor point:

Overall, the manuscript appears very brief. A more critical discussion of the data is required, with respect to limitations and virtues of the in vitro approach, as well as with respect to existing literature on the topic.

Reviewer #1:

In their paper "T cell stimuli independently sum to regulate an inherited clonal division fate," Marchingo et al study the nature and source of variation in clonal expansion of stimulated T cells. The main question they asked is why a population of (essentially) identical T cells displays heterogeneous division destiny(DD) (fig 1a)?

The authors proposed two scenarios that could lead to this phenomenon: 1) Clonal concordance (fig 1b top). Cells from the same clonal family have the same DD, but different clonal families may have different DD's. This implies that DD is lineage imprinted. 2) Clonal discordance (fig 1b bottom). Cells from the same clonal family have different DD's. In this case, DD could be either pre-programmed (e.g. asymmetric division) or stochastically regulated. The authors then developed an interesting multiplex-dye assay. The authors claim, based on their data (fig 2c) and model, that scenario 1 is favored.

At this point, the authors abruptly changed to another question: whether the signal integration at the clonal level was stochastically independent. Here, we do not see the connection of this question to their main question of clonal concordance.

We apologize for the terse structure of the paper you received originally. It was written to a short format and transferred without alteration to Nature Communications. With the additional space allowed by Nature Communications, we have expanded it substantially. In the revised introduction, we more systematically describe our hypothesis and aims, clarifying that the two questions of clonal concordance and single cell signal addition must be reconciled within the one 'correct' framework, which explains our desire to address both questions in the one paper. We believe these changes result in an important improvement in the piece's readability, and thank the reviewer for their feedback on this point.

We highlight the following new section at the end of the Introduction to illustrate these changes:

“Any clonal level answer to the question of relative concordance in DD must also be reconciled with a further striking population level observation: T cell DD is regulated by the type and the strength of the signals received, and many signal combinations result in both the means and variances of population DD distributions summing linearly, illustrated in Fig. 1c⁹. This suggests independence of the effects of signals driving DD. Thus the solution to the familial genesis of DD variation posed in Fig. 1a-b must also address how variable outcomes at single cell level result from fates of clonal family trees (Fig. 1d). We sought to identify the source of DD variation, and to determine how signal integration that is additive at the population level results from, and is consistent with, clonal family behaviour. To address these questions we developed and utilised a novel multiplex clonal division-tracking assay based on the combinatorial use of multiple division tracking dyes.”

In any case, the authors developed a mathematical framework and statistical test to support the claim that signal integration is independent. An interesting inference from this independence is "In particular, for signal integration to be independent at the clonal level, it appears necessary that receptor levels for different signals be uncorrelated in activated cells." However, the authors did not dig deeper into this direction.

We agree this is an important inference of our analysis, and one of the most significant of the study. The final results section of the paper addresses the prediction directly by measuring receptor levels and

sorting for differences and following effects on DD fate. Our results demonstrate independent assortment of the two signalling co-receptors on the surface of activated cells and reveal the significant consequences of the variation in CD28 expression level for directing inter-clonal DD fate differences. We have added further discussion to highlight the conclusion

Overall, there are some interesting ideas, assay and analytical framework are presented. However, we do not see a strong connection between the points they make based on the writing. Also, the authors did not dig deeply into their two claims: clonal concordance and independent signal integration. Finally, multiple assumptions in the model were not supported by data or discussion, which makes interpreting their results challenging. For these major claims, only one experiment is not sufficient. At this time, the study seems premature for publication.

In response to this comment, as well as those below where the additional material is explained in greater detail, we have expanded the description of all modeling assumptions. We have added primary data from experiments with FUCCI cells to substantiate our identification of quiescence (Methods & Fig. S2). We have added a new mathematical model and analyzed it to quantify the level of within-family concordance necessary to explain the clonal range data (Main Text, Methods, Fig 4c and Fig. S6c). We have added a repeat of the main experiment and performed the analysis of clonal concordance and signal addition on its data, resulting in identical conclusions (Fig S4, S6 and S7).

Below we provide some feedback on points of concern/confusion that we hope the authors will find constructive.

1. How can we know whether the observed clonal concordance is not simply due to the small number of divisions allowed before measurements were taken? A simulation of inherited vs. stochastically regulated division destinies would help determine how likely it is that the clonal concordance observed is truly due to an inherited property, as opposed to an effect of taking measurements after only a few divisions in which the cells are able to diverge. If the downstream analysis were limited to clones where at least half of the progeny were measured and all measured cells were quiescent, would the conclusion still hold?

The reviewer raises an important point with this comment. To tackle it we have built additional mathematical model, described in the main text and in supplementary methods, and fit it to the clonal range data (Fig. 4c and S6c). The model has one free parameter that encapsulates the per-generation within-family correlation of cells in DD fate. All other aspects of the model, including the cell sampling probability per-stimulation condition, are parameterized by the data. The resulting quantitative analysis strongly supports our original conclusion regarding high familial concordance. This is an important contribution to the study and we thank the reviewer for this suggestion.

2. Independent experiments are not combined in main figures nor are they given in the supplement. Only one experiment is performed per main claim, and alternative hypotheses are neither mentioned nor refuted.

A repeat of the main experiment, and of the analysis performed on it, can be found in Fig S4 and S6 of the revision. The results are consistent between it and that reported in the main text.

Expanding on the response to the previous point, we developed a mathematical model with a single parameter, ρ , that defines the pair-wise within-family correlation in DD fate per generation. With $\rho=0$ all cells within a family make entirely independent DD decisions. With $\rho=1$ all cells within a family cease dividing in the same generation. Irrespective of the value of that correlation, the

proportions of cells across families that progress to the next generation are, by design, consistent with each set of per-condition population data. In the model, cells from the resulting full family trees are sampled based on the experimentally determined proportion, capturing the incomplete nature of the experimental data.

For both the main experiment and the repeat in SI, we show data along with the output of the model for various values of ρ including a per-condition best fit value (Fig 4c & Fig S6c). Thus, while the original submission argued directly from observation for strong concordance over random assignment, the analysis in the revised paper allows us to more quantitatively reject the latter hypothesis and to quantify that the degree of correlation required to explain the data (>0.8). We appreciate the suggestion of the reviewer in moving in this quantitative direction.

3. We did not understand the connection between the investigation of clonal concordance and the independent additivity of signals.

As noted above, this connection is now explicitly addressed with expanded text and specific sections in the revised introduction, results and discussion.

4. Controls for quiescence: It is mentioned briefly that the gates chosen for quiescence were based on control cells, but this data is not shown. An accurate measure of quiescence is essential to the arguments made in the paper. Some experiments to demonstrate these cells are indeed quiescent may include inducing quiescence in the derived T cells to confirm that this size gating accurately segregates quiescent cells, or sorting out the cells within this gate and demonstrating that they have ceased dividing. The authors' previous work used FUCCI cells to determine gates for quiescence, but the mice in this manuscript do not appear to have FUCCI.

Our ability to identify quiescent cells is undoubtedly important for the study. The reviewer correctly notes that in previous work (Marchingo et al., Science, 2014) we used FUCCI cell cycle reporter T cells to identify non-cycling cells. In that paper, however, we also used the small cell gating that we employ here. The utility of this classification stems from the greater flexibility of not having fluorescent marked cells, while its appropriateness is a consequence of the significant correlation between cells identified with either method. To further substantiate this point, for the three time-points used in the main study, Figure S2 provides a representative primary data comparison of the output of the small cell gating strategy with one based on FUCCI expression. In all cases, the small cell gating gives a classification that is extremely consistent ($>95\%$) with the FUCCI approach, justifying its appropriateness. The revised manuscript also includes an expanded methods section to further explain this method of quiescence determination, as well as that additional data.

5. Why was 2.5 CFSE + 5 CTV removed from the analysis? This omission isn't clearly explained.

The multiplex method allows multiple configurations and up to ten unique family 'lanes'. In the specific experiment shown, it was determined from controls that there was too much overlap between the 2.5 + 5 group and others for them to be accurately separated, and hence they were not sorted into the wells. That led to the eight distinguishable configurations shown and has no bearing on the results or interpretation.

6. Intra-clone experiments are not repeatable, and cannot be assessed over time: Because of the nature of the method, there is no way to repeat experiments from the same clone. How do the authors conclusively distinguish between hypotheses such as genetic heterogeneity acquired over the lifetime of the mouse vs. the stochastic effects of recent T-cell stimuli?

While we are intent on identifying sources of variation in the original population, this query seems difficult to comment upon in the present paper. Our method traces considerable variation to the initial population, and at least some of that difference to stochastic differences in receptor numbers post initial stimulation. Determination of further influences of epigenetics would require additional sophisticated experiment design and would be a topic for future work as a consequence of the present findings.

7. Figure 3: Since the authors are comparing two distributions, why wasn't a KS or rank-sum test used, instead of making asymptotic assumptions in the modified chi-squared test?

This query makes apparent that our text was less clear than we intended. Given independent observations of X , Y , A and B , the null hypothesis is that the sum of two random variables, $X+Y$, is equal in distribution to the sum of two others, $A+B$. No existing statistical test is designed with this circumstance in mind, which is why we had to develop a new one based on existing results. As measurements of tree depth take discrete values, a convolutional multinomial chi-square test is the most appropriate thing to pursue, and there is a well-known general theory for the construction of tests of this sort, grounded in Moore's beautiful work in the '70s, which is what we employ.

The only approach by which one could use established tests without modification is statistically much weaker: one would truncate the X and Y data sets to the same size, and create the pairwise sums $X+Y$, then performing a similar operation on A and B , doing a multinomial test on the combined output. While this reduces the question to one that can be addressed with existing techniques, for these experiments it results in discarding a large volume of data in the process. Application of that approach to these data does not result in rejection of the null hypothesis, but as it is a much less stringent test of the null hypothesis than the one we created, ours is preferable.

As a direct comment on the two tests mentioned by the reviewer, the KS test and Wilcoxon rank-sum test, note that both assume a continuous underlying distribution, which is not the case for these data, and so would be inappropriate even if one discarded data as describe above. Moreover, both are based on the same asymptotic principles as chi-square tests, via the functional central limit theorem and asymptotic normality respectively, so that is not a distinguishing feature to their advantage anyway.

8. Variability of the initial labelling: In Figure 2, cells at t26 were selected based on "higher FSC fluorescence" and "being undivided" when sorted. Please clarify whether these cells were from the highest level of fluorescence observed (undivided). An experiment where cells are stained and immediately analyzed would establish the undivided fluorescence level. Additionally, a tight distribution of staining at this time would confirm that the stain intensity accurately reflects the number of divisions, and that variability in staining does not affect conclusions.

The time-point is chosen due to extensive published time course experiments indicating cells have undergone no more than one division at that time (Hommel and Hodgkin, *J. Immunol.*, 2007; Marchingo et al, *Science*, 2014), with the vast majority undivided and easily distinguished, and so we can assure the reviewer that we have sorted undivided cells. To clarify that cells were sorted as described in text, i.e. undivided cells are those with the highest dye fluorescence, the method diagram of the revised paper, Fig. 2 now includes FACS sort plots. We have also further clarified in the methods section that cells were sorted from a tight distribution of staining intensity in CTV and CFSE channels (i.e. in the dyes used for division distinction).

Reviewer #2:

In the manuscript entitled "T cell stimuli independently sum to regulate an inherited clonal division fate" the authors present a very elegant and stringently controlled assay for simultaneously tracking up to six cell divisions of multiple T cell clones harboring identical TCRs in vitro. Using this assay, they aim to decipher when and how the highly distinct clonal burst sizes, recently discovered to originate from single T cells responding to infection in vivo, are determined. The central question they pose is "whether clonal outcomes are intrinsically inherited from the initial cell or if they arise from stochastic extrinsic regulation" of division activity in the initial cell's progeny.

From their data the authors draw the following conclusions:

- 1) The division fate of a T cell clone is imprinted onto the initial cell and intrinsically inherited by its daughters.*
- 2) The relevant T cell stimuli received by the initial cell vary stochastically and independently from one another.*
- 3) Variation in clonal fate results from stochastic variation of costimulatory signal receptivity (before T cell activation) and variation in T cell receptor signal strength (during T cell activation).*

While appreciating the stringency of the well-controlled in vitro system provided by Duffy et al. this reviewer has some substantial concerns as to the validity of the far-reaching conclusions drawn from this dataset.

Main concerns:

- 1) The authors in vitro system (utilizing self-presentation, murine-IL-2 blocking antibody and addition of human-IL-2) deliberately limits the availability of extrinsic signals acting upon the individual members of an expanding T cell clone and thus, does not allow the conclusion that that these factors are generally irrelevant.*

We entirely agree with the reviewer. We have extended the introduction and discussion to make clear that in this study we have intentionally eliminated significant sources of extrinsic influence, as our purpose here is to understand the intrinsic programming. Extrinsic modifying environmental signals, such as a local hit of IL-2, would likely alter the fate of individual cells in a manner that could be disconnected from other family members, and we certainly do not conclude that extrinsic influences are generally irrelevant. Alternate experimental conditions, such as the inclusion of autocrine signaling or the supply of altered signals to developing families, are topics of future study. Indeed, we hope that with assays such as we provide here, along with in vivo tracking methods, the relative role of intrinsic and extrinsic signals for T cell responses can ultimately be identified.

As stated by the same group in Marchingo et al. Science 2014: "The strong effect of autocrine interleukin-2 (IL-2) was controlled by adding blocking antibody (clone S4B6) and using human IL-2 (hIL-2), resistant to S4B6, when required." Further experiments are required to resolve whether physiological availability of autocrine IL-2 increases intra-clonal variation.

This point is similar to the previous one. The question of extrinsic signal effects and autocrine cytokines such as IL-2 is now discussed in expanded discussion, but exploration of the impact of autocrine signaling on intra-clone variability is left as future work, either by us or by others utilizing the newly introduced assay in the present work.

Further on, the use of self-presentation limits the availability of diverse signals provided by professional APCs. In Extended Data Fig. 5 the authors show that prolonged presence of self-presenting cells in cultures does not lead to substantial intra-clonal variation. However, this could be due to the fact that

these non-professional antigen presenters do not provide the signals relevant for intra-clonal variation. The remaining question is, whether priming via professional APCs and/or their prolonged presence in culture will increase intra-clonal variation?

As presented, the fascinating feature of our study is that we have stripped away essentially all of the sources of extrinsic influence, and yet there is a remarkable level of inter-clonal heterogeneity, whose source we have traced to the experience of the original cell. That is, in our view, striking, and the primary message we wish the paper to communicate. The additional contribution of extrinsic stimuli on cell fate, as provided by APC or autocrine IL-2, is amenable to study in a similar manner, but is beyond the scope of our current study. The clear answer is that they can only increase the heterogeneity, but the surprising point we would like to emphasise is that APC and extrinsic IL-2, or other signals are in no way solely responsible for individual differences.

2) The authors' conclusion "that the summation of T-cell stimuli  effects on division number at the clonal level are stochastic and independent" could be substantially confounded by the specific setup of their experiments. In the authors' in vitro system, TCR stimuli are provided by self-presentation and co-stimulation is provided by soluble anti-CD28 antibody. Thereby, the authors' artificially detach factors from one another that are physiologically provided together - in the cellular context of the APC. Interdependence of TCR stimuli and CD28 signaling could well arise, if both signals were simultaneously provided by an APC. The physiological situation could look like this: An APC stochastically loaded with more peptide (higher pMHCI density) than another, will interact longer with a given T cell and thus provide more CD28-signaling via CD80/CD86 to this T cell (potentially overriding the effects of subtle variations of CD28 expression in naïve T cells). It thereby remains unresolved whether the use of professional APCs will abrogate the independent manner in which stimuli related to the TCR and to CD28 influence division fate?

Thank you for this feedback, which made apparent that we were unclear in the original text. This study investigates independence of costimulatory aCD28 and IL-2 effects, but the null hypothesis of independent additivity would continue to hold even if the expansion contribution of one or other of them, but not both, were directly dependent on the antigen signal. We have clarified this in the revised text. In the presence of antigen presenting cells, if TCR and aCD28 did indeed turn out to be interdependent it does not follow that this would alter independent additivity of this CD28 signal with IL-2. With expanded text now include discussion section on possible implications of the many differences in costimuli receptor levels versus experience of antigen.

3) The authors' conclusion that "clonal heterogeneity arises through differences in stochastic antigen interaction and initial receptor sensitivity" is only partially supported by their data. In support of their conclusion the authors show that sorted CD28-high/low and IL2Ra-high/low subpopulations will display distinct mean division fates. However, they do not investigate the impact of this intervention on variation of clonal division fate directly. Thus, they do not answer how much of the variability found in their system can actually be attributed to the suggested factors - and how much is left unexplained. In the opinion of this reviewer, the impact of reducing CD28 and IL2Ra variability, should best be investigated in a setting in which strong inter-clonal variability is detected to start out with (e.g. Fig. 2C / 72h + hIL2 + anti-CD28). The question would be, how much the variation of division fate between single clones can be homogenized if clones are preselected for defined criteria? This reviewer would suggest the following experiment: A Population of naïve OTI cells is sorted for CD28hi expression, then stimulated by self-presentation, then single cell-sorted at 26h for IL2Rahi and then monitored for 72h in media supplemented with hIL2.

We thank the reviewer for this suggestion. Supplementary Fig 10 in the revised article reports on a realization of a version of the proposed experiment, but as the existing data already shows that even without sorting on receptor levels one gets concordant clonal DD outcomes we reasoned that population data suffices to answer this point. The paper now reports that:

“To further investigate how much of the variability in DD can be attributed to receptor sensitivity we tested whether variation in T cell DD can be reduced if cells are preselected for defined receptor levels. OT-I/Bcl2l11^{-/-} CD8⁺ T cells were sorted for CD28^{hi} expression (top 20%), stimulated by N4 self-presentation, then single cell-sorted at 26h for IL-2R α ^{hi} (top 35%) and monitored for 72h in media supplemented with hIL2. The variation in division fate in these cells was compared to that of an unsorted population control. As shown in Supplementary Fig. 10, this sorting significantly reduced the variation in DD outcomes observed in the population, with a reduction in the population variance from 1.3 to 0.67.”

Minor point:

Overall, the manuscript appears very brief. A more critical discussion of the data is required, with respect to limitations and virtues of the in vitro approach, as well as with respect to existing literature on the topic.

We apologize for the terse version of the paper that the reviewer originally received.

REVIEWERS' COMMENTS:

Reviewer #1 (Remarks to the Author):

The authors have addressed our main concerns. The logical flow of the manuscript is much improved, clarifying many major points of confusion. Experimental replicates were performed as requested, and the additional analyses lend strength to the conclusions. Many decisions which were originally unclear (such as the statistical test) have been sufficiently explained both in the rebuttal and the manuscript itself. The new model supports the conclusion that division destinies are highly concordant and accounts for the incomplete capture of progeny. As such, we recommend acceptance of the manuscript.

Reviewer #2 (Remarks to the Author):

The manuscript has substantially improved. The scope of the study has been clarified. Its virtues, limitations and relevance for the field are now clearly discussed. The additional experiment (Fig. S9) now provides a better quantification of how much inter-clonal variation can be attributed to differences in CD28 expression and strength of the TCR signal (indirectly measured by aIL2R) and how much inter-clonal variation must be attributed to other factors.

The main findings of the paper are important and novel to the field.

Minor points:

Fig. S9 is indicated in the main text to be showing single cell data. As the authors indicate in their point-by-point response these data are actually generated by sorting populations of T cells. This error should be corrected.

Fig. S4 is not cited in the text.

Reviewer #1:

The authors have addressed our main concerns. The logical flow of the manuscript is much improved, clarifying many major points of confusion. Experimental replicates were performed as requested, and the additional analyses lend strength to the conclusions. Many decisions which were originally unclear (such as the statistical test) have been sufficiently explained both in the rebuttal and the manuscript itself. The new model supports the conclusion that division destinies are highly concordant and accounts for the incomplete capture of progeny. As such, we recommend acceptance of the manuscript.

Reviewer 1's remarks appear to require no response.

Reviewer #2:

The manuscript has substantially improved. The scope of the study has been clarified. Its virtues, limitations and relevance for the field are now clearly discussed. The additional experiment (Fig. S9) now provides a better quantification of how much inter-clonal variation can be attributed to differences in CD28 expression and strength of the TCR signal (indirectly measured by aIL2R) and how much inter-clonal variation must be attributed to other factors.

The main findings of the paper are important and novel to the field.

Minor points:

Fig. S9 is indicated in the main text to be showing single cell data. As the authors indicate in their point-by-point response these data are actually generated by sorting populations of T cells. This error should be corrected.

This oversight has been corrected.

Fig. S4 is not cited in the text.

This was originally only referenced only in the legend of Fig. 3, but is now referred to in the main text.